# A Study of Two Impactful Heavy Rainfall Events in the Southern Appalachian Mountains during Early 2020, Part I; Societal Impacts, Synoptic Overview, and Historical Context

**Douglas Miller** [1,*], **John Forsythe** [2], **Sheldon Kusselson** [2], **William Straka III** [3], **Jifu Yin** [4], **Xiwu Zhan** [5] and **Ralph Ferraro** [5]

1 Atmospheric Sciences Department, University of North Carolina Asheville, Asheville, NC 28804, USA
2 Cooperative Institute for Research in the Atmosphere, Colorado State University, Fort Collins, CO 80523, USA; John.Forsythe@colostate.edu (J.F.); Sheldon.Kusselson@colostate.edu (S.K.)
3 SSEC Cooperative Institute for Meteorological Satellite Studies, University of Wisconsin Madison, Madison, WI 53706, USA; william.straka@ssec.wisc.edu
4 ESSIC Cooperative Institute for Satellite Earth System Studies, University of Maryland, College Park, MD 20740, USA; jyin@umd.edu
5 NOAA-NESDIS Center for Satellite Applications and Research, College Park, MD 20740, USA; Xiwu.Zhan@noaa.gov (X.Z.); Ralph.R.Ferraro@noaa.gov (R.F.)
\* Correspondence: dmiller@unca.edu; Tel.: +1-828-232-5158

**Abstract:** Two heavy rainfall events occurring in early 2020 brought flooding, flash flooding, strong winds and tornadoes to the southern Appalachian Mountains. The atmospheric river-influenced events qualified as extreme (top 2.5%) rain events in the archives of two research-grade rain gauge networks located in two different river basins. The earlier event of 5–7 February 2020 was an event of longer duration that caused significant flooding in close proximity to the mountains and had the higher total accumulation observed by the two gauge networks, compared to the later event of 12–13 April 2020. However, its associated downstream flooding response and number of landslides (two) were muted compared to the April event (21). The purpose of this study is to understand differences in the surface response of the two events, primarily by examining the large-scale weather pattern and available space-based observations. Both storms were preceded by anticyclonic Rossby wave breaking events that led to a highly amplified 500 hPa wave during the February storm (a broad continent-wide 500 hPa cyclone during the April storm) in which the accompanying low-level cyclone moved slowly (rapidly). Model analyses and space-based water vapor observations of the two events indicated a deep sub-tropical moisture source during the February storm (converging sub-tropical low-level moisture streams and a dry mid-tropospheric layer during the April storm). Systematic differences of environmental stability were reflected in differences of storm-averaged rain rate intensity, with large-scale atmospheric structures favoring higher intensities during the April storm. Space-based observations of post-storm surface conditions suggested antecedent soil moisture conditioned by rainfall of the February event made the widespread triggering of landslides possible during the higher intensity rains of the April event, a period exceeding the 30 day lag explored in Miller et al. (2019).

**Keywords:** atmospheric rivers; extreme rainfall; landslides; southern Appalachian Mountains

## 1. Background

The challenge of observing and forecasting precipitation in mountainous regions of the mid-latitudes is well documented (e.g., references [1–18]). The latter challenge is inextricably linked to the former. Recent increased tourism and development in these regions has highlighted the shortcomings of these capabilities. To further complicate these issues, the challenges are dependent on the scale and season of the precipitation-generating storm. In the mid-latitudes, warm season storms tend to be of middle-to-small scales in

time and space, while cool season storms are much larger and longer-lived. The variety of storms experienced locally is dependent on the geographic location of the mountains. Mountains of the mid-latitudes located along the western boundary of continents generally exist in arid climates, such that the primary precipitation-producing storms occur in the cool season and are large in spatial and temporal scale. In contrast, mountains of the mid-latitudes located along the eastern boundary of continents exist in humid climates and have a greater variety of precipitation-producing storms. A cool season phenomenon common to both mountain regions in the east and west, often associated with significant rainfall events, is the atmospheric river (AR, [19–31]).

ARs are narrow and elongated zones of rapid, anomalously moist air at low levels originating from the sub-tropics and located just ahead of the surface cold front in mid-latitude cyclones, responsible for a significant portion of poleward vapor and latent heat transport (e.g., [32–35]). Mid-latitude cyclones and their associated ARs produce a higher percentage of the observed annual rainfall in regions along the western boundary of continents [36] than for regions of eastern continental boundaries [29], primarily due to differences in climate zone, ultimately, a function of the surface temperatures of nearby water bodies and subsequent background atmospheric water vapor content. ARs of western continental mid-latitude storms are also often responsible for catastrophic landslides in the coastal mountains, particularly following active wildfire seasons, when the exposed soil is vulnerable because of the loss of stabilizing vegetation root systems [37]. In eastern continental mid-latitude storms, the direct connection between the rainfall influenced by ARs and landslides is "muddy" [38] due to potential pre-conditioning of the soil by a variety of other rain-producing events.

The linkage between extreme (top 2.5%) rainfall events and landslides in the southern Appalachian Mountains was investigated by Miller et al. [38], who found that long periods of rainfall (extreme Elevated Rain Time Clusters (ERTCs)), often linked with individual extreme rainfall events (ExtR) and sometimes with ARs, showed a relatively high correlation with landslide days (Pearson correlation coefficient of 0.561 and $p$-value of 0.008 for 117 data pairs) occurring within 30 days of the ERTC termination. Of 46 extreme ERTC events detected in a 20+ year sample using a rain gauge network located in the southern Appalachians, 63% played a role in at least one landslide day. Other studies in the southern Appalachians [39] and elsewhere in the world [40,41] have shown time scales longer than a 30 day lag can have relevance to conditioning the soil for subsequent landslides. Water loss from a mid- or lower-soil (deep) layer, due to runoff and percolation, is slow because of low water velocity in porous soils. Soil conditions such as antecedent moisture content and soil type play a primary role in determining if positive pore water pressures during a rainfall event exceed some threshold, triggering a landslide. Because extreme ERTC events can last from days to months, the landslide forecast challenge lies in determining whether the long period of heavy rainfall is conditioning the soil for a slide that occurs during a subsequent precipitation event or if it is sufficient to serve as a trigger as the event is unfolding.

The purpose of this study is twofold; to examine the large-scale atmospheric and surface conditions (e.g., soil moisture, river gauge height) before, during, and immediately after two heavy rainfall events of early 2020 that contributed to landslides in the southern Appalachian Mountains and to examine the variability of the two events using an "up close" regional view with a focus on the operational nowcast utility of a variety of NOAA Geostationary Operational Environmental Satellite-R (GOES-R) and Joint Polar Satellite System (JPSS) atmospheric and surface products in evaluating the probability of the triggering of landslides in the mountains. One of the goals of this paper is to validate, or refute, the findings of Miller et al. [38], that extreme ERTCs, linked with ExtRs and ARs, are useful parameters for alerting forecasters and emergency managers to a high probability of the triggering of landslides. The validation datasets consist of gridded atmospheric model analyses, remote sensing products, documented landslides, and in situ observations of rainfall as recorded from rain gauge networks located in two river basins of the southern Appalachian Mountains.



## 2. Methodology

Observations from two rain gauge networks, archived atmospheric analyses of the Global Forecast System (GFS), space-borne observations of soil moisture, precipitable water, and flooding, and landslides in the southern Appalachian Mountains documented by the North Carolina Geological Survey (NCGS) serve as the primary data sets upon which are formulated the "big picture" conclusions of this study.

### 2.1. Surface-Based Observations

Rainfall observations of two rain gauge networks located in the Pigeon River Basin (PRB, Figure 1, Table A1) and the Coweeta Sub-River Basin (CRB, Figure 1, Table A1) of the southern Appalachian Mountains, known hereafter as the Duke Great Smoky Mountains Rain Gauge Network (Duke GSMRGN) and the Coweeta Hydrologic Laboratory Rain Gauge Network (CHLRGN), serve as the reference data sets for defining event severity in early 2020. Observations from the Duke GSMRGN and CHLRGN used in this study have been collected for 11 and 86 years, respectively, and were utilized in Miller et al. [38] to investigate potential links between ARs, ExtR, and ERTC events and landslides. Observations of the Duke GSMRGN are used primarily to illuminate the variability of the two impactful events investigated in this study as the 32 rain gauges are located at elevations varying from 1036 to 2003 m, covering the PRB area (1823 km$^2$). The elevation range of the PRB within Haywood County, North Carolina varies from 427 m, where the Pigeon River enters Tennessee, to 2018 m (the top of Mt. Guyot), the fourth highest peak east of the Mississippi River. Terrain slopes are quite high near the mountains (57% near Mt. Guyot [42]) and flatten substantially in terraces and flood plains utilized for agriculture. Space-based estimates of land use and cover in the PRB vary from deciduous forest (64%), pasture (12%), evergreen forest (7%), low intensity residential (6%), to mixed forest (5%) [43]. After forest, land use and cover by vegetation type is dominated by cropland, followed in decreasing proportion by shrubland, savanna, and grassland [44]. For convenience, a listing of abbreviations unique to the study are included in Table A2 of Appendix B.

Observations of the 86-year-old CHLRGN provide primarily a historical context of the two events as the nine rain gauges cover a smaller elevation range (687 to 1366 m) and area (16.3 km$^2$) compared to the Duke GSMRGN. The elevation range of the CRB within Macon County, North Carolina varies from 675 to 1592 m [45]. As in the PRB, terrain slopes are also quite high through most of the CRB (56% near the upper Camprock Branch [42]). The Coweeta Hydrologic Laboratory, of the U.S. Department of Agriculture, has served as a climate station since 1934 for experimenting with forest management practices and monitoring climate change in the CRB [45]. Hence, land use and cover categories of the CRB have changed little from the forests of 1934, with the exception of the experimental sub-watersheds. The close proximity of the CRB to the Blue Ridge Escarpment (BRE, Figure 1) allows for the investigation of a potential enhancement of rainfall observed during the two events under favorable wind conditions.

Following the methodology of Miller et al. [46], total rainfall accumulation observed by the two rain gauge networks was binned into synoptic 6 h periods (0000, 0600, 1200, and 1800 Coordinated Universal Time (UTC) corresponding to the 6 h time resolution of the Global Forecast System (GFS) historical analysis National Centers for Environmental Information (NCEI) archives. Events were defined as having concluded when no amounts were recorded at any of the network gauges during at least a single synoptic 6 h period [47]. Non-zero per gauge accumulation amounts of each consecutive synoptic 6 h period were added to calculate the event total per gauge accumulation. The method for defining individual Elevated Rain Time Cluster (ERTC) events was similar to that for defining individual rainfall events described above, except that an ERTC event was defined to have ended when a 30 h period occurred without a single rain gauge detecting rainfall, the median rain-pause period documented by Miller et al. [38] based on the 80+ year CHLRGN archive.

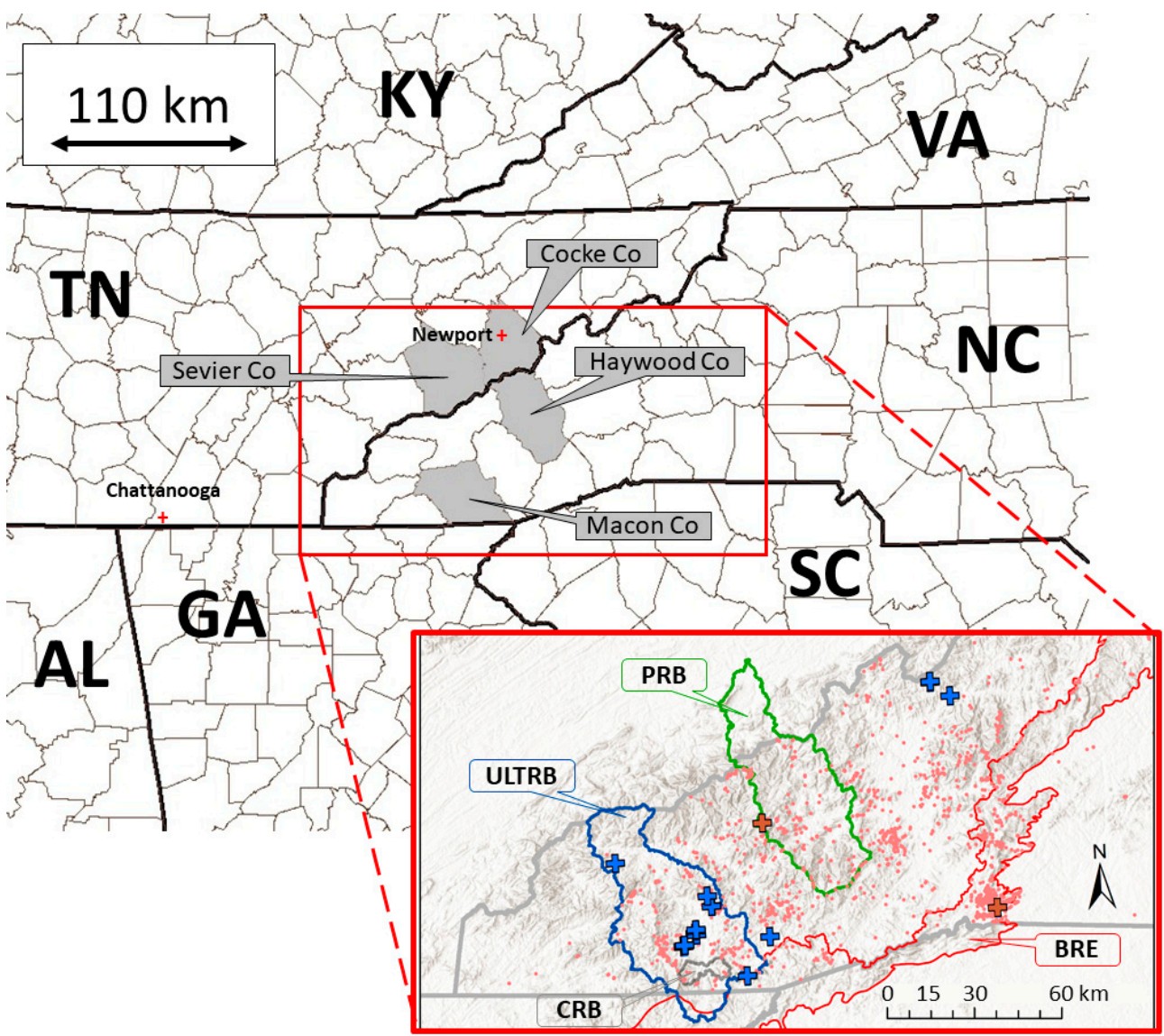

**Figure 1.** Locations of the Pigeon River Basin (PRB, green outline) and Coweeta sub-River Basin (CRB, gray outline), a sub-basin of the Upper Little Tennessee River Basin (ULTRB, blue outline), and topography (shaded) of the southern Appalachian Mountains. The Pigeon River Basin (PRB) corresponds to the borders of Haywood County, North Carolina and extends northward slightly into Cocke and Sevier Counties, Tennessee. The Coweeta River Basin (CRB) is located in Macon County, North Carolina. Specifics on the locations and elevations of individual rain gauges of the Duke GSMRGN, located in the North Carolina region of the PRB, and CHLRGN, located in the CRB, are provided in Table A1. The center points of the PRB and CRB are located 60 km apart. The Blue Ridge Escarpment (labeled "BRE" and outlined in red) is the boundary between the Blue Ridge and the Piedmont physiographic province. The brown (blue) color-filled "+" symbols highlight two (21) landslide locations documented by the NCGS initiated by the 5–7 February (12–13 April) 2020 heavy rainfall event. Coral dots highlight landslide locations initiated since 1940 not occurring in February or April 2020. Locations of Newport and Chattanooga, Tennessee are also highlighted with a red "+" symbol.

Landslide inventory data for North Carolina used in the study came from the landslide geodatabase maintained by the NCGS [48]. The geodatabase documents 23 landslides of various types for the February–April 2020 focus period of this study (color-filled "+" symbols in Figure 1), where the known date(s) of movement for individual landslides are recorded in the geodatabase.

### 2.2. Atmospheric River Detection Algorithm

The process for flagging the presence of ARs during the study period is described in Miller et al. [46]. The vertically integrated horizontal water vapor transport (IVT; [49]) was computed from the 0.5° NCEI GFS historical analysis grids via the expression

$$-\int_{po}^{p}(q\boldsymbol{V})\frac{\mathrm{d}p}{g},\tag{1}$$

where $q$ is the specific humidity, $\boldsymbol{V}$ is the horizontal wind, $p_o$ is 1000 hPa, $p$ is 100 hPa, and $g$ is the acceleration due to gravity. GFS-based analysis IVT fields were examined closely over the IVT study domain, a 10° longitude by 5° latitude region centered in longitude on 83°W, and directly south of, the center of the PRB (35.5° N, 83° W, *c.f.* Figure 2 of Miller et al. [46]). The AR detection algorithm searched for IVT features located within the IVT study domain and required that the feature influenced the domain for at least 12 h with a mean IVT of at least 500 kg m$^{-1}$ s$^{-1}$. A minimum duration of eight hours was required in the studies of Neiman et al. [50,51] and Ralph et al. [52] along the U.S. west coast. The more conservative time restriction of 12 h was selected because GFS-analysis fields were only available every six hours.

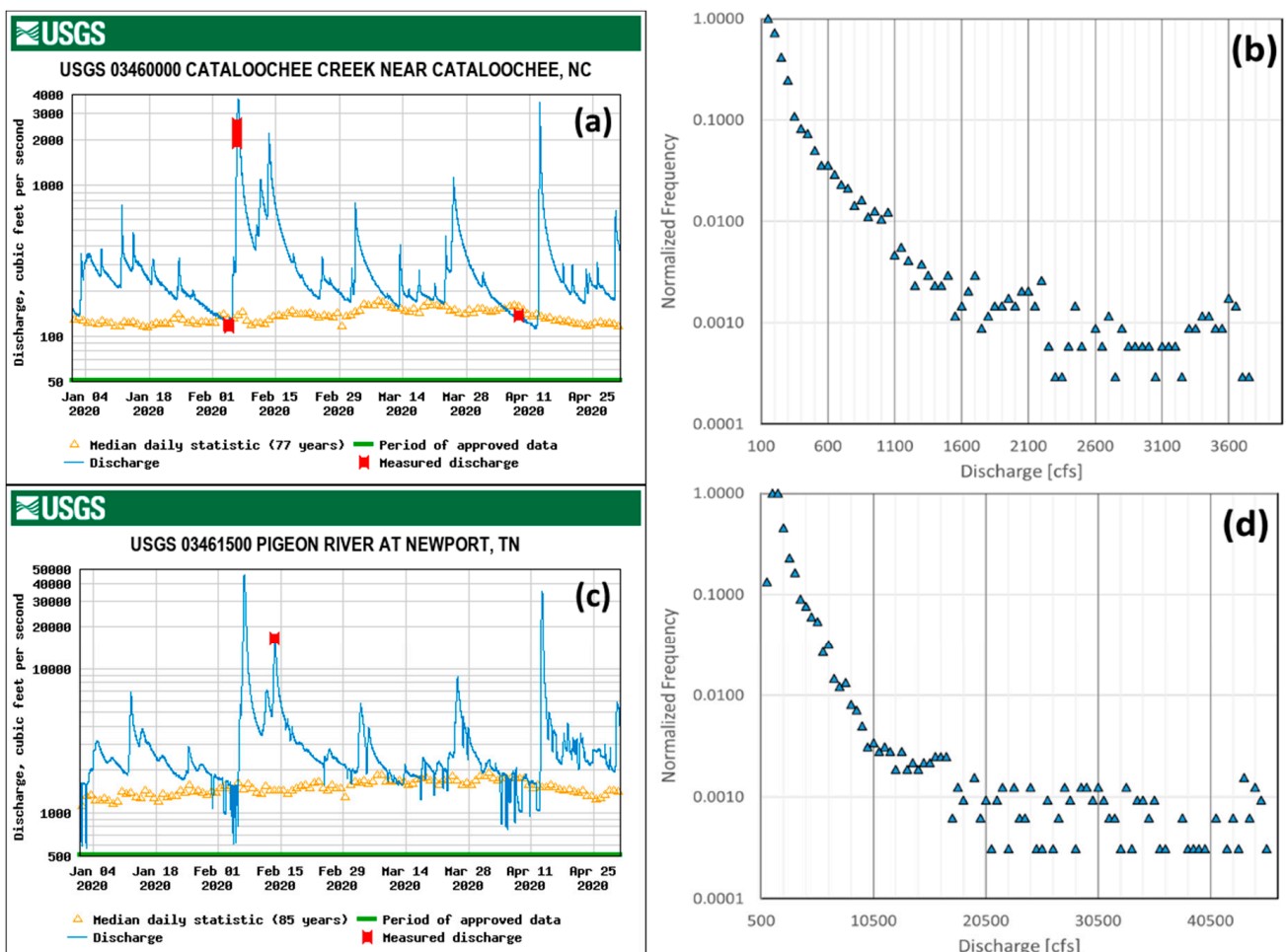

**Figure 2.** USGS discharge (cfs) hydrographs (**a**) at Cataloochee Creek near Cataloochee, North Carolina, and (**c**) at the Pigeon River at Newport, Tennessee over the 1 January–30 April 2020 period and corresponding histograms (**b**,**d**) (accessed online https://waterdata.usgs.gov/nc/nwis/rt and https://waterdata.usgs.gov/tn/nwis/rt, accessed on 5 June 2021). Histogram bin size for panel (**b**,**d**) is 50 and 500 cfs, respectively.

### 2.3. Space-Borne Observations

2.3.1. Soil Moisture

Soil moisture is the water content of the soil layer as the water balance of precipitation, evapotranspiration, runoff and infiltration. A significant precipitation event could increase the soil moisture content dramatically. Since the soil moisture content is directly related to the soil dielectric constant that impacts the microwave emission or reflectance of the land surface, satellite sensors observing low frequency microwave emission or reflection are thus capable of detecting the soil moisture changes. Currently there are several satellites in space providing soil moisture (SM) sensitive observations. A Soil Moisture Operational System (SMOPS) has been developed at the National Oceanic and Atmospheric Administration (NOAA) National Environmental Satellite, Data, and Information Service (NESDIS) to produce a one-stop upper-layer SM shop from all currently available microwave satellite sensors since 2012 [53–55]. The SMOPS can uniquely provide near real time (NRT) blended SM data on the global domain to satisfy the users' requirements of real time global SM datasets [54,55]. The latest SMOPS V3.0 (hereafter: SMOPS) was operationally released in 2017 with combining microwave SM retrievals including the Soil Moisture Active and Passive (SMAP, [56]), Soil Moisture and Ocean Salinity (SMOS, [57]), the Advanced Microwave Scanning Radiometer (AMSR)-2 onboard the Global Change Observation Mission-Water (GCOM-W) satellite [58] and the Advanced Scatterometer (ASCAT) both from Meteorological Operational Platform (MetOp)-A and MetOp-B satellites [53–56,59,60]. The SMOPS V3.0 has higher accuracy, more reasonable spatial pattern and higher data availability than the older versions [55,61]. Compared to the individual satellite soil moisture data products, the SMOPS blended product offers high spatial coverage, and reliable and continuous long-term upper-layer soil moisture datasets [53,60,62]. The operational gridded SMOPS product was available at a 0.25° latitude and longitude horizontal resolution [55].

2.3.2. Precipitable Water

Passive microwave satellite sounding instruments allow the broad profile of water vapor to be retrieved in cloud, non-precipitating conditions [63]. Layer precipitable water (LPW) between two selected pressure levels can be derived from these soundings via integration of the mixing ratio profile [64]. Model winds were used to advect the satellite microwave water vapor retrievals to a common time, creating a product called Advected LPW (ALPW) [65]. ALPW is created hourly at CIRA from seven polar orbiting spacecraft and depicts atmospheric moisture in four layers (surface–850, 850–700, 700–500, 500–300 hPa). ALPW provides forecasters something they had been missing from multi-satellite Total Precipitable Water; a satellite depiction of the vertical structure of water vapor in clear and cloudy areas. A key feature of ALPW is that it is independent of the forecast model water vapor fields, so it allows forecasters to assess how well the models are handling water vapor.

2.3.3. Flooding

The basis for flood detection from the Visible Infrared Imaging Radiometer Suite (VIIRS) on GOES-R and the Advanced Baseline Imager (ABI) on the Suomi National Polar Orbiting Partnership (S-NPP) satellite and the Joint Polar Satellite System (JPSS) series of NOAA polar-orbiting meteorological satellites are the spectral characteristics in the VIIRS and ABI visible (VIS), near infrared (NIR), and short-wave infrared (SWIR) channels. A comprehensive introduction to the VIIRS flood algorithm can be found in Li et al. [66], with a similar algorithm applied to the ABI instrument. The VIIRS flood algorithm and products have been extensively validated and evaluated with satellite imagery and ground based measurements [66], and are currently in the process of being operationalized. A water reference map at normal conditions was used to identify flooding water and marked as water percentage in a pixel from 1% to 100%. The daily VIIRS products take the flood maps from each granule for both NOAA-20 and S-NPP, and were composited into the maximum daily, clear sky, flood extent. Cloud clearing was performed through a maximal

water fraction composition process. The daily ABI composite was carried out in a similar manner, with the rolling compositing ABI Flood Products based on each of the 10-min ABI flood maps. At the end of each day, the Joint Daily VIIRS/ABI product blended the daily flood composite from VIIRS and used the 1-km ABI daily clear-sky flood extents to fill in the gaps of clouds and cloud shadows from the VIIRS product.

## 3. Results

The 6 h synoptic periods covering observed rainfall of the Duke GSMRGN of the two events in early 2020 spanned the periods 0600 UTC 5 February–1200 UTC 7 February (54 h) and 1200 UTC 12 April–1800 UTC 13 April 2020 (30 h). Several societal hazards were experienced in connection with the events, including flooding, flash flooding, strong thunderstorm winds, and tornadoes during each event, and flooding and landslides either during or shortly after each event.

### 3.1. Societal Hazards
#### 3.1.1. Flooding

A well-traveled road in the Great Smoky Mountains National Park for elk sightseeing located next to Cataloochee Creek (in Haywood County) was closed to visitors in the spring of 2020 due to substantial damage resulting from flooding associated with the heavy rainfall of the 5–7 February and 12–13 April 2020 events. The United States Geological Survey (USGS) river gauge at Cataloochee Creek recorded a discharge rate exceeding 1000 cubic feet per second (cfs) over a 36 h period from 0900 UTC 6 February to 2100 UTC 7 February 2020, with a peak of 3760 cfs (Figure 2a,b). The 76-year median discharge rate at this gauge in February is nearly 150 cfs. The second event at the creek recorded a discharge rate exceeding 1000 cfs over a 14.5 h period from 0545 UTC–2015 UTC 13 April 2020, with a peak of 3580 cfs. The period of substantial discharge during the 5–7 February event was over twice as long as that of the 12–13 April event. A flooding event at Cataloochee Creek is considered extreme if the discharge rate exceeds 3000 cfs, making the road of a nearby campground impassible and justifying its evacuation [67]. Although the two heavy rainfall events of early 2020 were not unusual, occurring, on average, every two to five years, "The loss of large hemlock and other trees to disease and nonnative pests" was responsible for a number of "downed trees causing major erosion issues during the storm events" [67]. This problem has increased noticeably over the last five years [67].

Further downstream from Cataloochee Creek, along the Pigeon River, at the USGS river gauge in Newport, Tennessee (Figure 1), the National Weather Service (NWS) has determined the major flood stage at this location of the river to be 12 feet (3.66 m). The river gauge recorded a major flood over a 20.5 h period at Newport from 1700 UTC 6 February–1330 UTC 7 February 2020, with a peak gauge height of 18.2 feet (5.55 m) and a peak discharge rate of 45,500 cfs (Figure 2c,d). The second event at the river recorded a major flood over a 12.25 h period at Newport from 1015 UTC–2230 UTC 13 April 2020, with a peak gauge height of 16.43 feet (5.01 m) and a peak discharge rate of 35,400 cfs. The period of major flooding at Newport during the 5–7 February event was just under twice as long as that of the 12–13 April event. From a historical perspective, the peak crests of the February and April 2020 heavy rainfall events ranked third and sixth, respectively, over the period of record at Newport, Tennessee starting in 1901 [68]. Yet further downstream of Newport, along the Tennessee River, at the Tennessee Valley Authority gauge at Chattanooga, Tennessee (Figure 1) the flooding associated with the two events was significant, but not historic [68]. The severity of river discharge associated with the two heavy rainfall events was found to be comparable just downstream of the CRB (Little Tennessee River, USGS #3503000, not shown) and the river discharge of the April event was more significant than the February event upstream of Newport, in the southern PRB (Pigeon River, USGS #3456991, not shown), and downstream of Newport, near Chattanooga, Tennessee (South Chickamauga Creek, USGS #3567500, not shown).

Using estimates of regional flooding as observed by VIIRS/ABI over a rectangular area downstream of the Tennessee–North Carolina border, extending from Newport (northeast corner) to the Chattanooga region (southwest corner), the downstream effects were the reverse of expectations based on peak river discharge observations of Figure 2a,c. Although the skies cleared at midday on 7 February, daily flooded pixels were not detected in the Newport–Chattanooga region until 8 February (26 of 60,641 observable pixels; 0.04%) and peaked in number on 9 February (1723 of 104,126 observable pixels; 1.65%) and dropped rapidly (less than 0.1%) thereafter. In contrast, after the skies cleared early on 13 April, detection of daily flooded pixels occurred almost immediately on 13 April (269 of 50,003 observable pixels; 0.54%) and remained elevated for several days (14 April; 2203 of 66,098 observable pixels; 3.33%, 15 April; 3518 of 85,132 observable pixels; 4.13%, 16 April; 3283 of 133,118 observable pixels; and 2.47% and 17 April; 2474 of 58,893 observable pixels; 4.20%). Even though the February event was of longer duration in terms of observed rainfall (54 h) in the PRB and CRB, the detected downstream effects were relatively brief in time and severity. The April event was of shorter duration regarding observed rainfall (30 h), but the detected downstream effects were prolonged and severe. This is consistent with what was observed regarding landslide responses of each event, which will be described in the next section.

Another regional perspective of the societal hazards associated with the two heavy rainfall events early in 2020 is gleaned from storm reports as documented by the NWS (Table 1). During the February 2020 event, over 75% of the reports originated from the North and South Carolina side of the mountains (Greer, South Carolina NWS forecast office (GSP)), with a majority related to reports of flooding and flash flooding. Note that all convection-related reports (thunderstorm wind and tornado) during the February event came from south and east of the primary spine of the southern Appalachian Mountains. In contrast to the February event, reports associated with the April event were primarily convection-related and distributed on both sides of the spine of the mountains. In terms of the number of reports, the April event was more active north and west of the mountains (Morristown, Tennessee NWS forecast office (MRX)) than the February event, particularly significant since the April event duration was 24 h shorter in duration than the February event.

**Table 1.** Number of official storm reports documented by the National Weather Service (NWS) during the 5–7 February and 12–13 April 2020 heavy rainfall events within the County Warning Area (CWA) of the Greer, SC office (GSP) and the Morristown, TN office (MRX).

| Event | 5–7 February 2020 | | 12–13 April 2020 | |
|---|---|---|---|---|
| NWS CWA | GSP | MRX | GSP | MRX |
| Flood | 29 (38%) | 22 (96%) | 2 (4%) | 0 |
| Flash Flood | 32 (42%) | 1 (4%) | 7 (15%) | 14 (45%) |
| Thunderstorm Wind | 6 (8%) | 0 | 26 (57%) | 11 (36%) |
| Tornado | 9 (12%) | 0 | 11 (24%) | 6 (19%) |
| **Total** | **76** | **23** | **46** | **31** |

3.1.2. Landslides

Landslides surveyed in North Carolina by personnel of the North Carolina Geological Survey (NCGS) triggered by the two heavy rainfall events in early 2020 (color-filled "+" symbols on the inset of Figure 1) found two landslides associated with the February 2020 event and 21 with the April 2020 event. One of the two initiated in February 2020 occurred within the PRB, on its southwest boundary near Bunches Bald. Most of the 21 events initiated by the April 2020 event occurred within the southern half of the Upper Little Tennessee River Basin (ULTRB), near the CRB. A number of landslides from April 2020 occurred in close proximity to each other just north of the CRB and are indistinct in Figure 1 due to the scale of the map inset. NCGS surveys of the events [48] indicated all were soil-driven (shallow), with rupture depths ranging from 0.91 to 4.57 m. Of the 17 documented slope configurations at the initiation points, nine were initiated on unmodified slopes. The

event initiated near Bunches Bald was on a modified slope and covered a slide area of just over 0.4 acres (undocumented rupture depth). The two most noteworthy landslides of early 2020 occurred after the April event near the CRB and consisted of landslides initiated on unmodified slopes. The first slide encompassed 1.4 acres (length of 366 m, rupture depth of 3.0 m) and appeared to fail as a result of both groundwater discharge and surface water runoff from the heavy rainfall associated with the April event. The most significant slide, considered a major event, covered an area of 16.6 acres (length of 1158 m, undocumented rupture depth) and was estimated to move at a speed exceeding 5 m s$^{-1}$. Its failure appeared to occur in response to significant amounts of soil-bedrock (groundwater) seepage. Eyewitness accounts of six of the April slides observed initiation early in the morning of 13 April 2020. The initiation time of the Bunches Bald slide was undocumented.

### 3.2. Synoptic Setting

"Hang-back" troughs [69] at the 500 hPa level, often resembling the anticyclonic Rossby wave breaking scenarios identified in Moore et al. [70] and Bosart et al. [71], have been associated with extreme precipitation events in the southern Appalachians (Miller et al. [38,46]) and are sometimes linked with two ARs (c.f., case events ranked 2, 7, and 8 in Table 4 of Miller et al. [46]). The cut-off low at the 500 hPa level located in the southwestern U.S. slows the propagation of the large-scale wave so the anticyclone centered offshore of the southeastern U.S. has ample time to humidify a significant portion of the lower troposphere as air streams spiral clockwise about the anticyclone, originating from the tropics and/or sub-tropics. Air flow at low levels associated with a jet stream or streak aloft located downstream of the hang-back trough transports the humid air rapidly into the southern Appalachians and the lift provided by the strong large-scale dynamics downstream of the large-scale trough, supplemented by orographic lift, results in significant rainfall amounts.

### 3.2.1. GFS Analyses

The 500 hPa level hang-back trough scenario preceded both the February 2020 (Figure 3a) and April 2020 event (not shown). In addition to a 500 hPa level cut-off low at 0000 UTC 4 February 2020 (Figure 3a), air streams of high mixing ratios (and equivalent potential temperature, $\theta e$) were established in western Tennessee at the 600 and 850 hPa levels (Figure 3b,d), associated with the circulation of an elongated 850 hPa level trough, extending southwestward from a local geopotential height minimum located over Lake Michigan (Figure 3c,d). These humid air streams marked the position of a mature AR that gradually dissipated over the next 24 h period (Figure 4). By 0000 UTC 5 February 2020, cyclonic vorticity advection downstream of the absolute vorticity maximum in the base of the 500 hPa level trough (Figure 4a) contributed to the cyclogenesis of a 850 hPa level storm located over northern Mexico (Figure 4c,d) that would become the dominant weather producer of the February 2020 event. This developing storm and its circulation in the lower troposphere spawned the formation of a second AR over the Gulf of Mexico at 1200 UTC 5 February 2020 (not shown) that started making its way into the Gulf States at 850 hPa by 0000 UTC 6 February 2020 (Figure 5d). At this time, the dominant storm at the 850 hPa level was centered over central Illinois (Figure 5c) and the 500 hPa level trough had pivoted substantially so that it was almost neutrally tilted (Figure 5a). Over the next 12 h, the associated AR matured to its greatest magnitude and horizontal extent (Figure 6a) as the 850 hPa level storm continued to deepen. By 0000 UTC 7 February 2020, the 850 hPa level cyclone center had propagated to a location between Lakes Erie and Ontario (Figure 7c,d) and the axis of the humid air streams associated with the AR (Figure 7b,d) was located along the U.S. east coast. The 500 hPa level trough by now had pivoted to a negatively tilted orientation (Figure 7a).

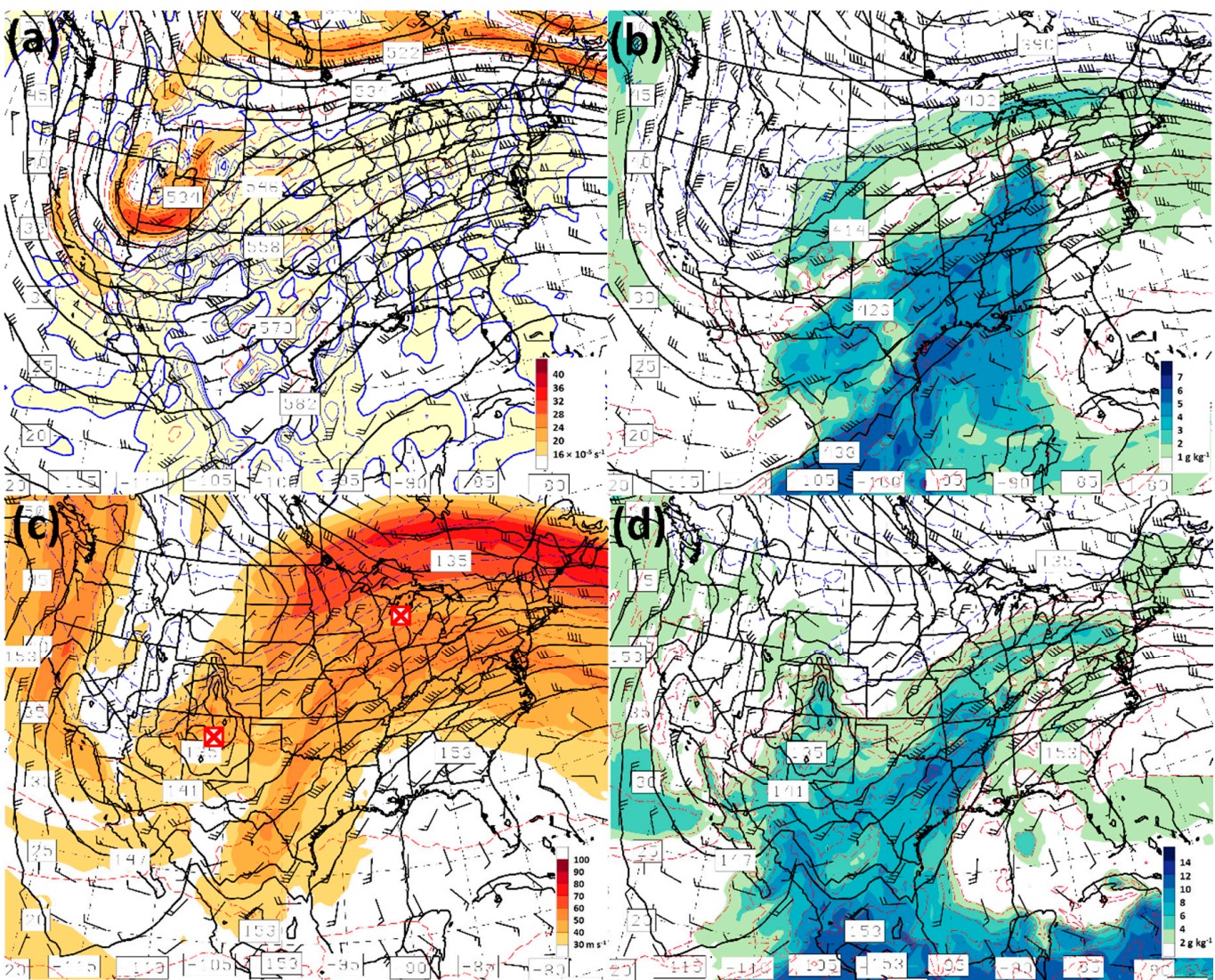

**Figure 3.** GFS−analyzed fields valid at 0000 UTC 4 February 2020 of: (**a**) 500 hPa level geopotential height (dam, solid black contours), absolute vorticity × 10 $^{-5}$ (s$^{-1}$, shading and red dashed contours), and wind vectors (kt) and 700 hPa level rising motion × 10$^{-3}$ (hPa s$^{-1}$, vanilla shading and blue solid contours); (**b**) 600 hPa level geopotential height (dam, solid black contours), mixing ratio (g kg$^{-1}$, shading), equivalent potential temperature (K, final blue (first red) dashed contour value is 304 K (310 K)), and wind vectors (kt); (**c**) 850 hPa level geopotential height (dam, solid black contours) and wind vectors (kt) and 1000-500 hPa layer thickness (dam, final blue (first red) dashed contour value is 540 dam (546 dam)), and 300 hPa level wind speed (m s$^{-1}$, shading); and (**d**) 850 hPa level geopotential height (dam, solid black contours), mixing ratio (g kg$^{-1}$, shading), wind vectors (kt), and equivalent potential temperature (K, final blue (first red) dashed contour value is 280 K (286 K)). Red cross-square symbols in panel (**c**) represent the locally dominant 850 hPa level cyclone center.

A hang-back trough was also associated with the April 2020 heavy rainfall event, but preceded the event in the southern Appalachians long enough beforehand that its role was to pre-condition (humidify) the lower tropospheric air streams and not contribute directly to the development of the associated 850 hPa level cyclone. By the time of the onset of precipitation in the PRB and CRB, the associated AR was not yet fully developed at 1200 UTC 12 April 2020 (Figure 8b,d), when the 850 hPa level cyclone center was located over the Kansas–Oklahoma border (near its panhandle) and in north-central Kansas (Figure 8c,d). Strong cyclonic vorticity advection downstream of the 500 hPa level absolute vorticity lobe over southwestern Kansas (Figure 8a) resulted in strong cyclogenesis of the low-level cyclone. By 0600 UTC 13 April 2020, the AR was fully formed (Figure 6b)

as the 850 hPa cyclone continued substantial development until its propagation over eastern Lake Superior at 1200 UTC 13 April 2020 (Figure 9c,d). The 600 hPa air stream had relatively low mixing ratio values (Figure 9b) in the air stream associated with the AR so that mid-tropospheric horizontal vapor transport was rather modest. As will be shown in a follow-on paper, this vertical differential of water vapor transport between the 600 and 850 hPa level (Figure 9b,d) contributed to potential convective instability (decreasing $\theta e$ with height) within broad swaths of the AR. Strong 700 hPa level ascent along a band extending from central North Carolina, through South Carolina, into southeast Georgia (Figure 9a) provided the trigger to release the instability, resulting in societal hazards related to convective processes; flash flooding, thunderstorm winds, and tornadoes.

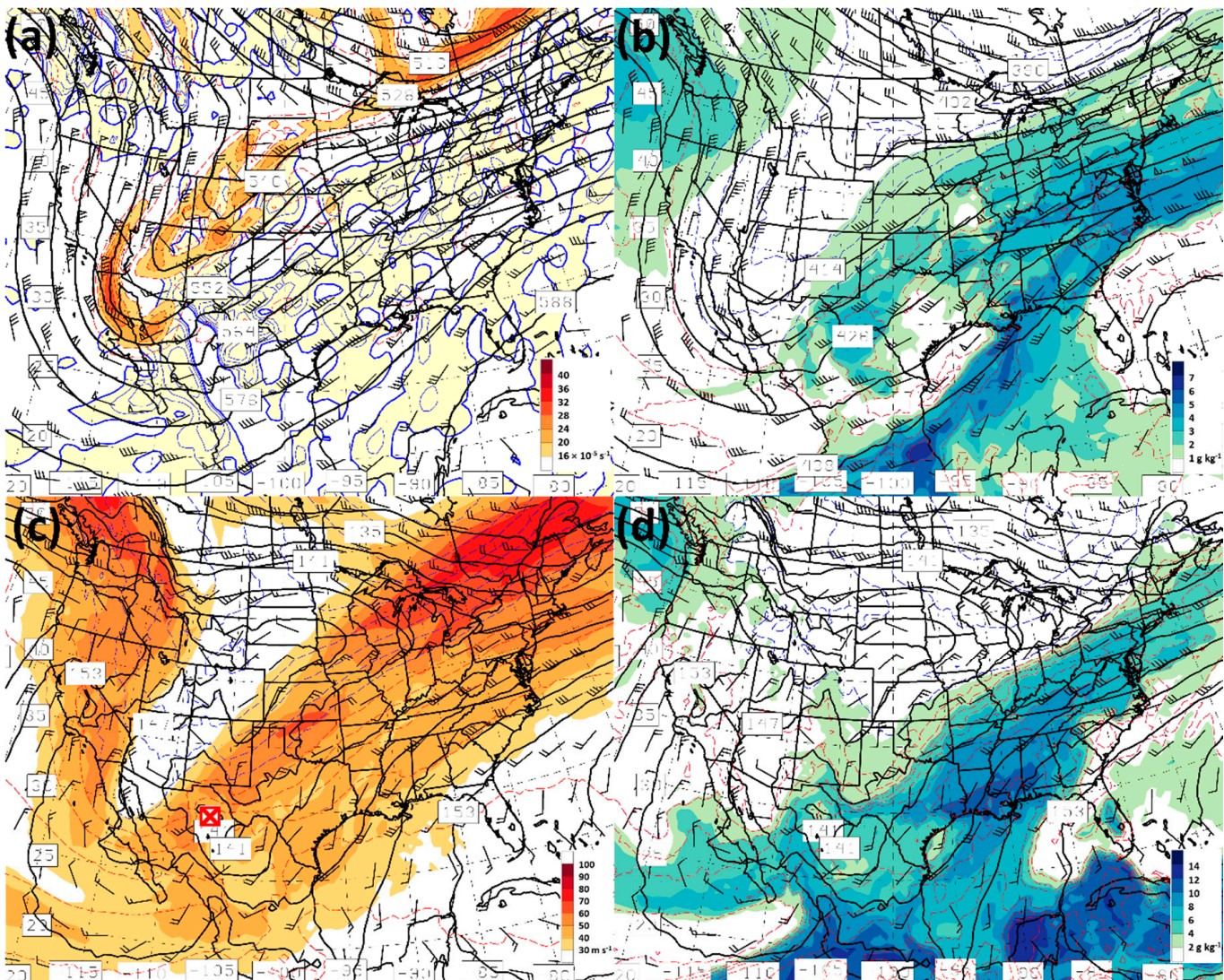

**Figure 4.** As in Figure 3, except GFS−analyzed fields valid at 0000 UTC 5 February 2020 of: (**a**) 500 hPa level geopotential height (dam, solid black contours), absolute vorticity $\times\ 10^{-5}$ (s$^{-1}$, shading and red dashed contours), and wind vectors (kt) and 700 hPa level rising motion $\times 10^{-3}$ (hPa s$^{-1}$, vanilla shading and blue solid contours); (**b**) 600 hPa level geopotential height (dam, solid black contours), mixing ratio (g kg$^{-1}$, shading), equivalent potential temperature (K, final blue (first red) dashed contour value is 304 K (310 K)), and wind vectors (kt); (**c**) 850 hPa level geopotential height (dam, solid black contours) and wind vectors (kt) and 1000-500 hPa layer thickness (dam, final blue (first red) dashed contour value is 540 dam (546 dam)), and 300 hPa level wind speed (m s$^{-1}$, shading); and (**d**) 850 hPa level geopotential height (dam, solid black contours), mixing ratio (g kg$^{-1}$, shading), wind vectors (kt), and equivalent potential temperature (K, final blue (first red) dashed contour value is 280 K (286 K)). Red cross-square symbol in panel (**c**) represents the locally dominant 850 hPa level cyclone center.

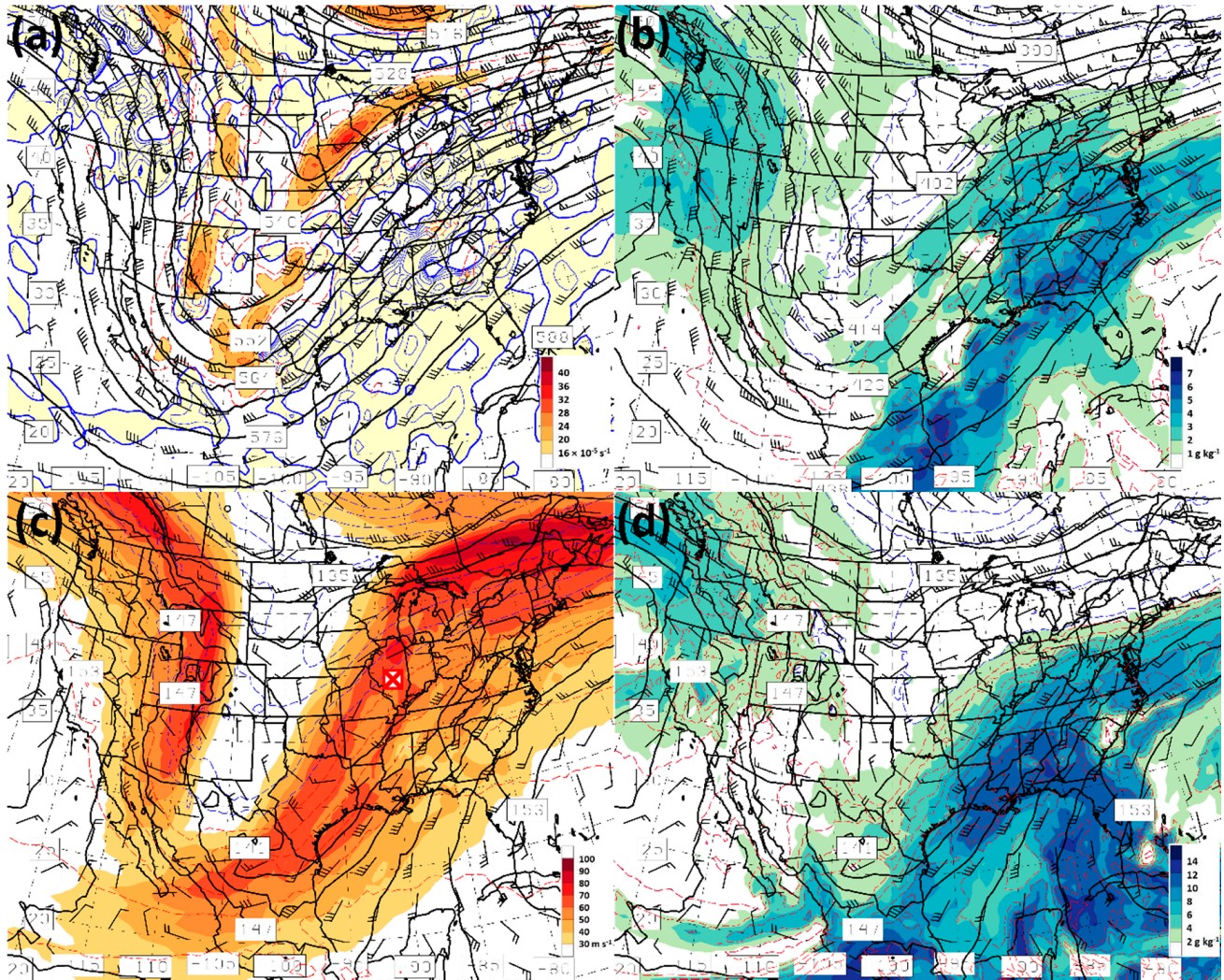

**Figure 5.** As in Figure 3, except GFS−analyzed fields valid at 0000 UTC 6 February 2020 of: (**a**) 500 hPa level geopotential height (dam, solid black contours), absolute vorticity $\times\ 10^{-5}$ ($s^{-1}$, shading and red dashed contours), and wind vectors (kt) and 700 hPa level rising motion $\times\ 10^{-3}$ (hPa $s^{-1}$, vanilla shading and blue solid contours); (**b**) 600 hPa level geopotential height (dam, solid black contours), mixing ratio (g $kg^{-1}$, shading), equivalent potential temperature (K, final blue (first red) dashed contour value is 304 K (310 K)), and wind vectors (kt); (**c**) 850 hPa level geopotential height (dam, solid black contours) and wind vectors (kt) and 1000–500 hPa layer thickness (dam, final blue (first red) dashed contour value is 540 dam (546 dam)), and 300 hPa level wind speed (m $s^{-1}$, shading); and (**d**) 850 hPa level geopotential height (dam, solid black contours), mixing ratio (g $kg^{-1}$, shading), wind vectors (kt), and equivalent potential temperature (K, final blue (first red) dashed contour value is 280 K (286 K)). Red cross-square symbol in panel (**c**) represents the locally dominant 850 hPa level cyclone center.

Although the two events have some similarities (e.g., AR-influenced, heavy rainfall producers), their origin and evolution were quite distinct. The high-amplitude 500 hPa trough of the February 2020 event essentially pivoted about a point in southern Canada as it transitioned from a positive- to a slight negative-tilt orientation during the 5–7 February 2020 period (Figures 3a, 4a, 5a and 7a). At the surface, the southeastern U.S. was on the warm side of a slow-moving low-level trough/baroclinic zone as the upper wave pivoted through the central U.S. (Figures 3d, 4d, 5d and 7d). By the time the dominant low-level cyclone made its way into the northeastern U.S. (Figure 7c), it was still undergoing vigorous development. In contrast, the positively tilted 500 hPa trough axis early in the April 2020 event (Figure 8a) was gradually absorbed by a very large-scale cyclone that dominated

North America (Figure 9a). The dominant low-level cyclone of 12–13 April 2020 rotated as a shortwave rather rapidly about the large-scale 500 hPa cyclone (Figures 8c and 9c) and tracked 1200 km to the west of the low-level Feb 2020 cyclone as it approached the north-central U.S. (Figure 9c).

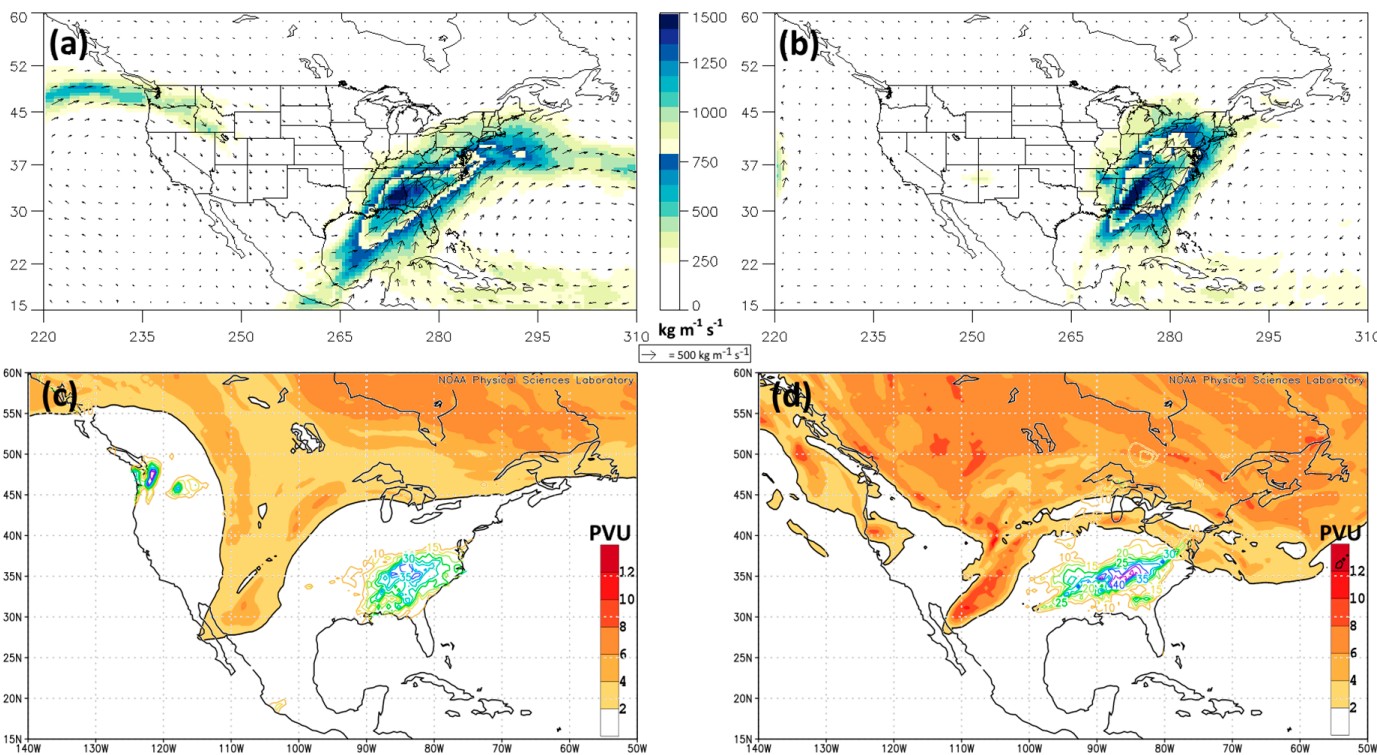

**Figure 6.** GFS−analyzed fields of IVT (kg m$^{-1}$ s$^{-1}$) valid at: (**a**) 1200 UTC 6 February 2020, and (**b**) 0600 UTC 13 April 2020. Reference IVT vector is plotted under the IVT color bar. GFS-analyzed fields of isentropic potential vorticity at: (**c**) 1200 UTC 5 February 2020 on the 315 K surface, and at (**d**) 0600 UTC 12 April 2020 on the 325 K surface. Color bar in panels (**c,d**) are in PV Units (PVU), where= 1 PVU = 10$^{-6}$ K kg$^{-1}$ m$^2$ s$^{-1}$. Shading is for 2 PVU or greater, where the 2 PVU (thick black) contour corresponds to the intersection location of the dynamic tropopause with the isentropic surface. Panels (**c,d**) also contain contours of daily mean precipitation accumulation (mm) for the: (**c**) 72 h, and (**d**) 48 h period commencing 5 February and 12 April 2020, respectively. Daily mean precipitation contours start at 10 mm, with an interval of 5 mm. Maximum daily mean precipitation contour for panel (**c,d**) is 40 and 50 mm, respectively. Precipitation plots created using a web page maintained by the NOAA Physical Sciences Laboratory based on Climate Prediction Center Global Unified Gauge-Based Analysis of Daily Precipitation data (accessed online https://psl.noaa.gov/data/gridded/data.cpc.globalprecip.html, accessed on 3 June 2021).

Both events represented examples of anticyclonic Rossby wave breaking that led to heavy rainfall events as highlighted in Moore et al. [70]. Following the PV detection methodology of Moore et al. [70], isentropic potential vorticity (PV) streamers (Figure 6c,d) having orientations consistent with anticyclonic Rossby wave breaking were found at similar upstream locations 24 h before the subsequent storm-influencing AR was centered on western North Carolina and the southern Appalachian Mountains (Figure 6a,b). Although the position and orientation of the PV streamers preceding the two events was similar, there were significant differences in their shape and PVU strength, primarily a consequence of the streamers being defined on different isentropic surfaces. Some of the shape and intensity differences were due to differences in the interaction of the polar front and subtropical jets (Figures 5c and 8c) during each event. Differences were also due to changes in season, as zonal mean isentropic surfaces retreat poleward from winter (February) to spring (April). The high amplitude (meridional) shape of the February 2020 500 hPa level wave pattern led to a slow-moving (long duration) event as the low-level cyclone traveled poleward downstream of the 500 hPa trough. Its confinement in the zonal direction led to event pre-

cipitation occurring in a relatively narrow corridor at and east of the southern Appalachian Mountains (Figure 6c). In contrast, the large-scale cyclone-dominant 500 hPa pattern of the April 2020 event guided the low-level cyclone across a broader expanse in the zonal direction as it traveled poleward, influencing a larger region with its precipitation centered on the southern Appalachians (Figure 6d). The geographic distribution of official NWS storm reports (Table 1) reflected differences in the 500 hPa wave structure and evolution of the two events.

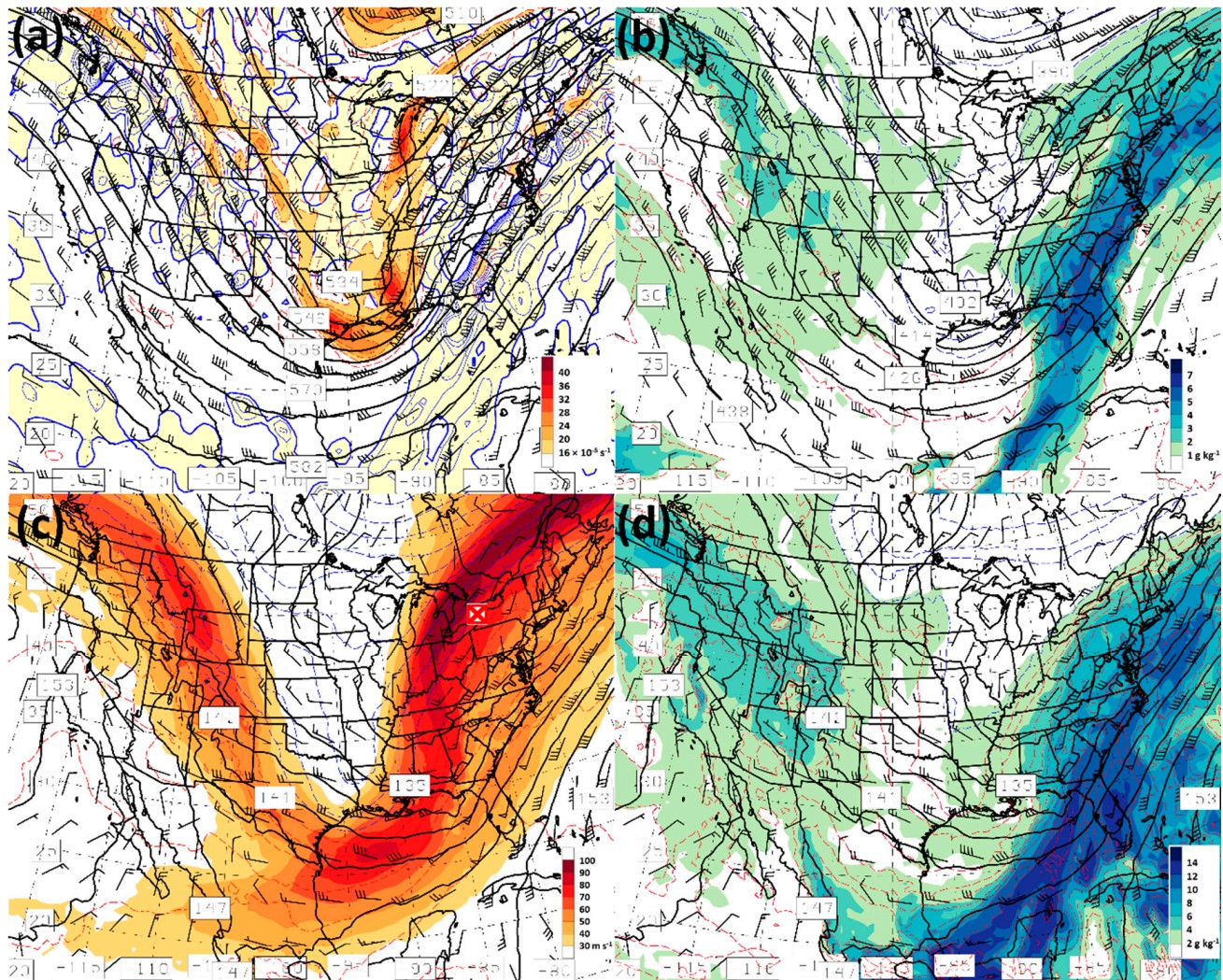

**Figure 7.** As in Figure 3, except GFS−analyzed fields valid at 0000 UTC 7 February 2020 of: (**a**) 500 hPa level geopotential height (dam, solid black contours), absolute vorticity $\times 10^{-5}$ (s$^{-1}$, shading and red dashed contours), and wind vectors (kt) and 700 hPa level rising motion $\times 10^{-3}$ (hPa s$^{-1}$, vanilla shading and blue solid contours); (**b**) 600 hPa level geopotential height (dam, solid black contours), mixing ratio (g kg$^{-1}$, shading), equivalent potential temperature (K, final blue (first red) dashed contour value is 304 K (310 K)), and wind vectors (kt); (**c**) 850 hPa level geopotential height (dam, solid black contours) and wind vectors (kt) and 1000-500 hPa layer thickness (dam, final blue (first red) dashed contour value is 540 dam (546 dam)), and 300 hPa level wind speed (m s$^{-1}$, shading); and (**d**) 850 hPa level geopotential height (dam, solid black contours), mixing ratio (g kg$^{-1}$, shading), wind vectors (kt), and equivalent potential temperature (K, final blue (first red) dashed contour value is 280 K (286 K)). Red cross-square symbol in panel (**c**) represents the locally dominant 850 hPa level cyclone center.

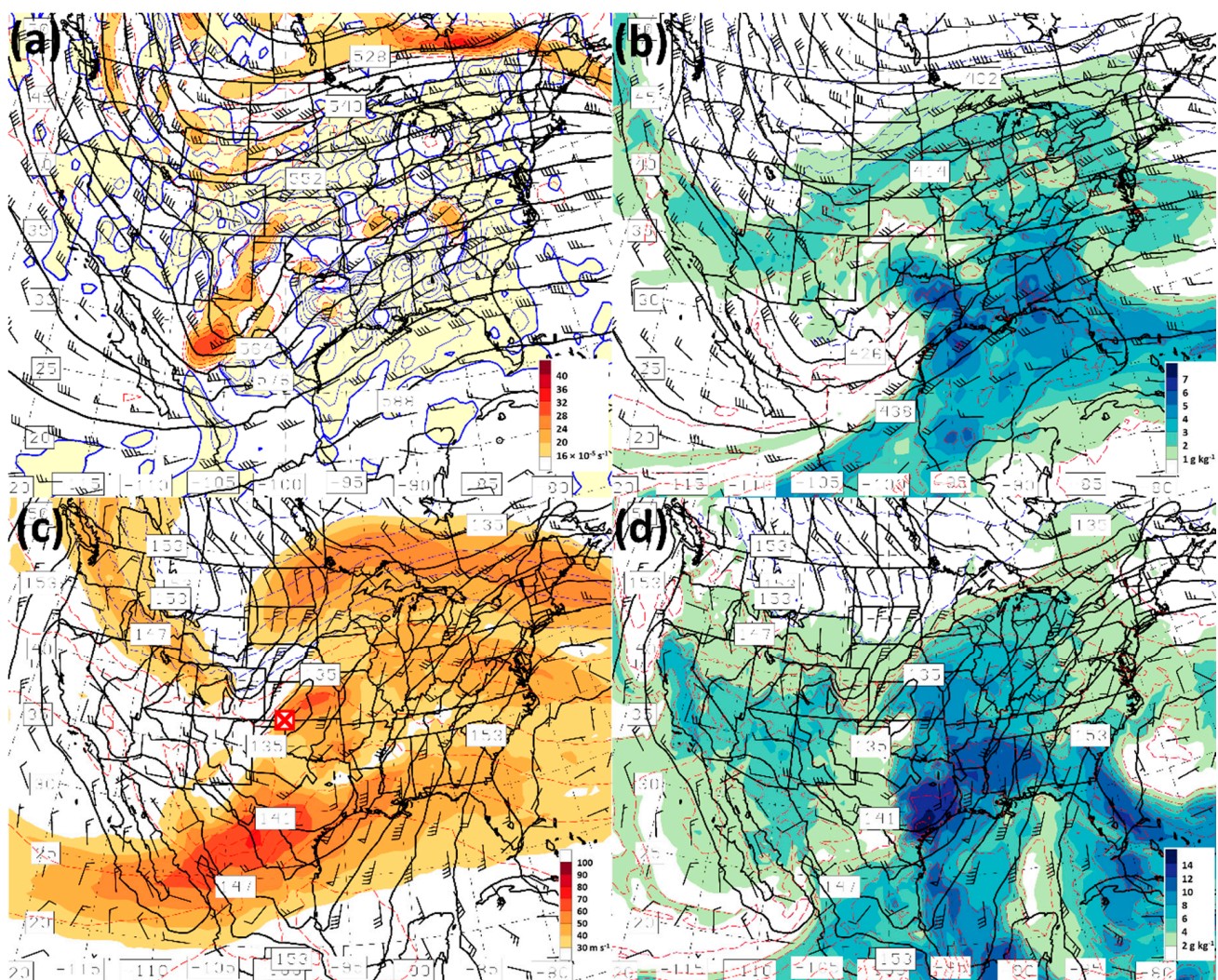

**Figure 8.** As in Figure 3, except GFS−analyzed fields valid at 1200 UTC 12 April 2020 of: (**a**) 500 hPa level geopotential height (dam, solid black contours), absolute vorticity $\times 10^{-5}$ ($s^{-1}$, shading and red dashed contours), and wind vectors (kt) and 700 hPa level rising motion $\times 10^{-3}$ (hPa $s^{-1}$, vanilla shading and blue solid contours); (**b**) 600 hPa level geopotential height (dam, solid black contours), mixing ratio (g $kg^{-1}$, shading), equivalent potential temperature (K, final blue (first red) dashed contour value is 304 K (310 K)), and wind vectors (kt); (**c**) 850 hPa level geopotential height (dam, solid black contours) and wind vectors (kt) and 1000-500 hPa layer thickness (dam, final blue (first red) dashed contour value is 540 dam (546 dam)), and 300 hPa level wind speed (m $s^{-1}$, shading); and (**d**) 850 hPa level geopotential height (dam, solid black contours), mixing ratio (g $kg^{-1}$, shading), wind vectors (kt), and equivalent potential temperature (K, final blue (first red) dashed contour value is 280 K (286 K)). Red cross-square symbol in panel (**c**) represents the locally dominant 850 hPa level cyclone center.

Examination of dynamic factors contributing to the strength of ascent downstream of the 500 hPa trough or cyclone yielded insignificant differences between the two events. Neither warm air advection nor cyclonic vorticity advection (not shown) yielded differences that could account for the significant difference in the strength of ascent highlighted by the GFS analyses near the southern Appalachian Mountains at the 700 hPa level (Figures 7a and 9a). The 12–13 April 2020 event demonstrated significantly stronger upward motion within the humid southerly flow compared with that of the February 2020 event. A comparison of $\theta e$ at the 600 and 800 hPa levels in both events (Figure 7b,d and Figure 9b,d) showed pockets of convective neutrality or instability (decrease in $\theta e$ with height) in the warm sector of the April 2020 storm that were of limited extent during the February event. Hence, given a fixed amount of dynamical forcing, a stronger vertical

motion response occurs in an atmosphere with weaker stratification. Stratification differences of the two events will be explored in detail in a follow-on paper to this study. Given differences in the strength of ascent (Figures 7a and 9a) and similarities in the horizontal transport of water vapor (IVT, Figure 6a,b) one would expect higher intensity rain rates associated with the 12–13 April 2020 event.

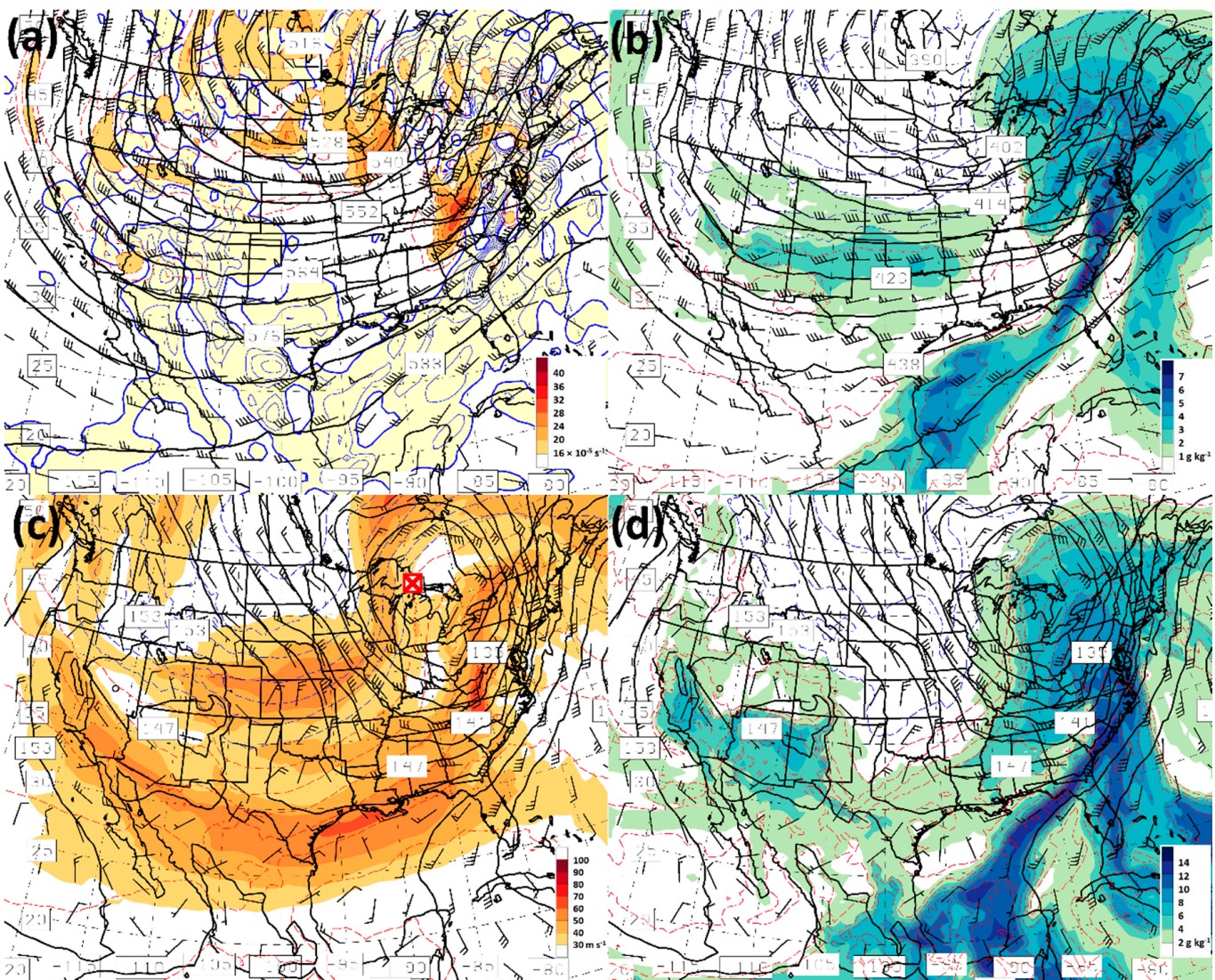

**Figure 9.** As in Figure 3, except GFS−analyzed fields valid at 1200 UTC 13 April 2020 of: (**a**) 500 hPa level geopotential height (dam, solid black contours), absolute vorticity $\times$ $10^{-5}$ (s$^{-1}$, shading and red dashed contours), and wind vectors (kt) and 700 hPa level rising motion $\times$ $10^{-3}$ (hPa s$^{-1}$, vanilla shading and blue solid contours); (**b**) 600 hPa level geopotential height (dam, solid black contours), mixing ratio (g kg$^{-1}$, shading), equivalent potential temperature (K, final blue (first red) dashed contour value is 304 K (310 K)), and wind vectors (kt); (**c**) 850 hPa level geopotential height (dam, solid black contours) and wind vectors (kt) and 1000-500 hPa layer thickness (dam, final blue (first red) dashed contour value is 540 dam (546 dam)), and 300 hPa level wind speed (m s$^{-1}$, shading); and (**d**) 850 hPa level geopotential height (dam, solid black contours), mixing ratio (g kg$^{-1}$, shading), wind vectors (kt), and equivalent potential temperature (K, final blue (first red) dashed contour value is 280 K (286 K)). Red cross-square symbol in panel (**c**) represents the locally dominant 850 hPa level cyclone center.

### 3.2.2. Soil Moisture

An examination of antecedent soil moisture conditions in the southern Appalachian Mountains is an important consideration for understanding differences in the number of landslides triggered by each of the heavy rainfall events early in 2020. Daily 25 km NOAA

Climate Prediction Center morphing method (CMORPH) precipitation accumulation [72] showed heavy rainfall in a domain centered on the southern Appalachian Mountains (34.25° N, 85.00° W to 37.25° N, 80.00° W) on 6 February 2020 (90+ mm, Figure 10) and a secondary daily accumulated rainfall peak on 13 April 2020 (40+ mm). SMOPS-based domain-averaged daily upper-soil layer moisture estimates (red curve in Figure 10) demonstrate the variability of shallow soil moisture observed over the first 120 days of 2020 (up until 29 April 2020), ranging from 0.34 $m^3 m^{-3}$ in early January to 0.26 $m^3 m^{-3}$ in mid-January 2020. In addition, in the SMOPS shallow soil moisture time series (Figure 10), the observed drop-off in post-event shallow soil moisture is worthy of note, with a minimum being reached two (three) days after the conclusion of the February (April) 2020 rainfall event. In situ point measurements of shallow (20 cm) soil moisture in the PRB by instrumentation of the North Carolina Environment and Climate Observing Network [73], supported by the State Climate Office of North Carolina, showed moisture ranges of 0.43–0.59 and 0.26–0.37 $m^3 m^{-3}$ at low (WAYN) and high (FRYI) elevation locations, respectively, over the same 120-day period. Shallow moisture was highly variable over the period due to drastic changes in precipitation, winds and temperature; the final two factors being primary drivers of evapotranspiration at the surface.

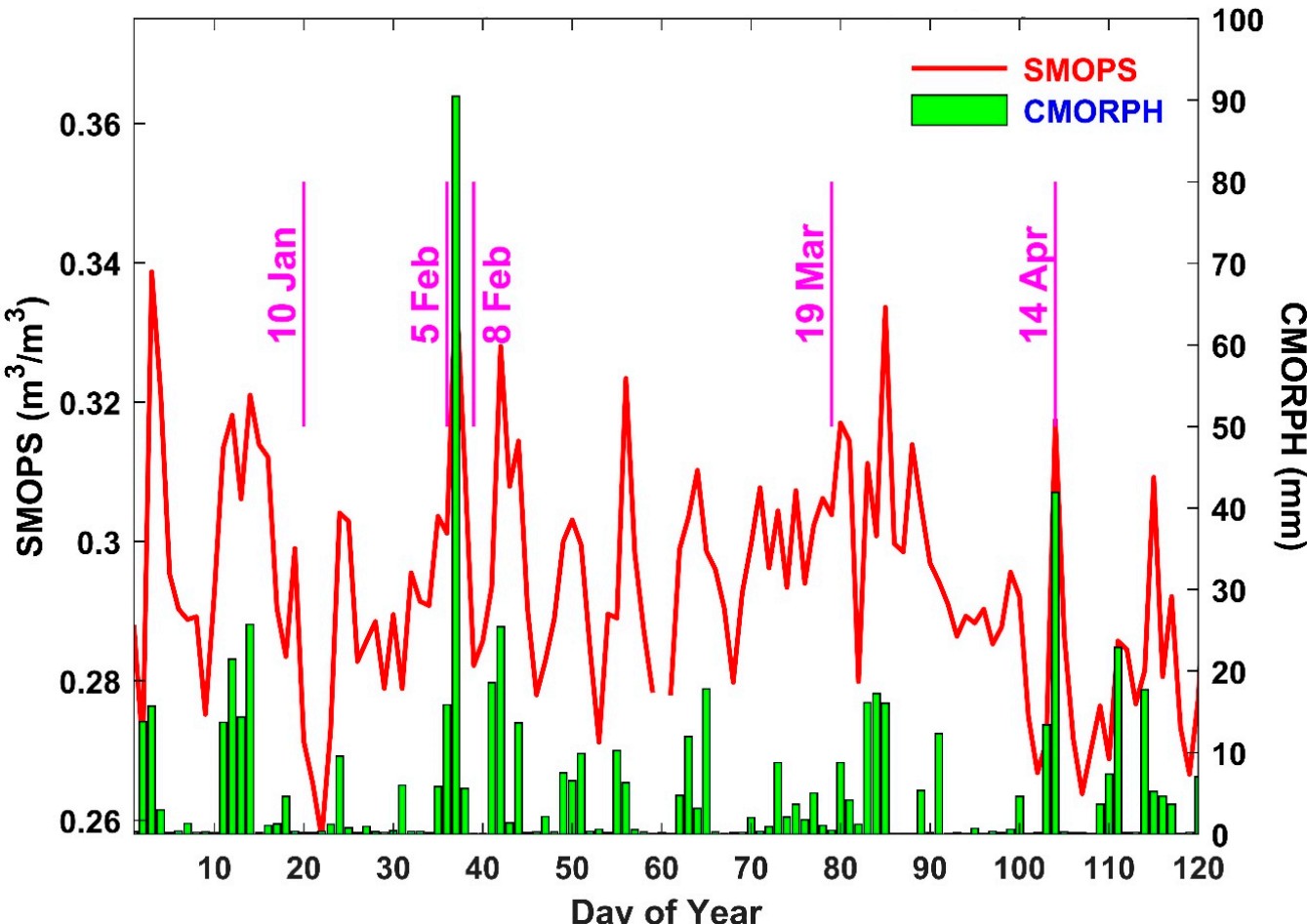

**Figure 10.** Average of daily SMOPS and CMORPH over the 1 January (day 1)–29 April 2020 (day 120) period for the domain extending from 34.25°N, 85.00°W to 37.25°N, 80.00°W.

Focusing on upper-layer soil moisture conditions within a focused study domain centered on the PRB (35.30°N, 83.30°W to 35.80°N, 82.80°W) and CRB (35.00°N, 83.50°W to 35.10°N, 83.40°W), the same general conclusions as the broader domain are reached regarding antecedent shallow soil moisture. SMOPS-based domain-averaged daily soil

moisture in the PRB (CRB) just before the February and April 2020 events was 0.31 and 0.29 (0.27 and 0.23) $m^3 \, m^{-3}$, respectively. These antecedent moisture conditions might suggest the soil was better primed for landslides just ahead of the February 2020 event. However, NCGS surveys of the 23 landslides of early 2020 showed a minimum rupture depth of 0.91 m, below the observable soil layer of SMOPS. For those landslides triggered primarily by groundwater discharge, SMOPS observations have limited applicability. The observed post-event shallow soil moisture drop-off of the focused PRB and CRB study domains was also steeper after the February 2020 event in both small domains. Of note regarding shallow soil moisture, 12–13 April was far enough into the spring season in the southern Appalachians that leaves of trees at lower elevations have opened and can contribute an additional draw-down of soil moisture via evapotranspiration. However, this effect should have hastened the observed soil moisture drop-off after the April 2020 event, contrary to what was indicated by SMOPS.

### 3.2.3. Blended Precipitable Water

Regional views of ALPW for the two cases are shown in Figures 11 and 12. In Figure 11, the surface–850 hPa and 850–700 hPa layers are shown, while the 700–500 and 500–300 hPa layers are shown in Figure 12. There were similarities and differences in the vertical water vapor structure of both events. In the lower troposphere, both cases tapped a water vapor source from the Gulf of Mexico and the tropical Atlantic. The surface–850 hPa layer values were up to 25 mm. The April case had higher surface–500 hPa layer vapor content originating from the western tropical Atlantic Ocean (Figure 11b,d and Figure 12d), but the February event had a deeper moist layer than the April event, with slightly higher values above 700 hPa (Figure 12c) over the Gulf of Mexico and eastern tropical Pacific Ocean. Differences in moisture content of the 700–500 hPa layer between the February (higher) and April (lower) event over the southeastern U.S. were reflective of differences in the 850–700 hPa level convective instability, with the latter event having a greater potential of instability. Storm reports implied that the April event was more convective than the February event. Analyses of 850–700 hPa level equivalent potential temperature in a follow-on paper of this study confirm the less stable environment of the April event and, combined with large-scale lift present downstream of the 500 hPa cyclone, provided a sufficient trigger of convective potential.

In Figure 12 (above 700 hPa), a deep connection to the tropical eastern Pacific across Mexico was apparent, and the April case had an additional stream of moisture from the sub-tropical Atlantic which was not present in the February event. Convergence of multiple pipelines of water vapor at different layers has been detected in advance of past flash flood events [65] as they provide additional fuel to support heavy precipitation.

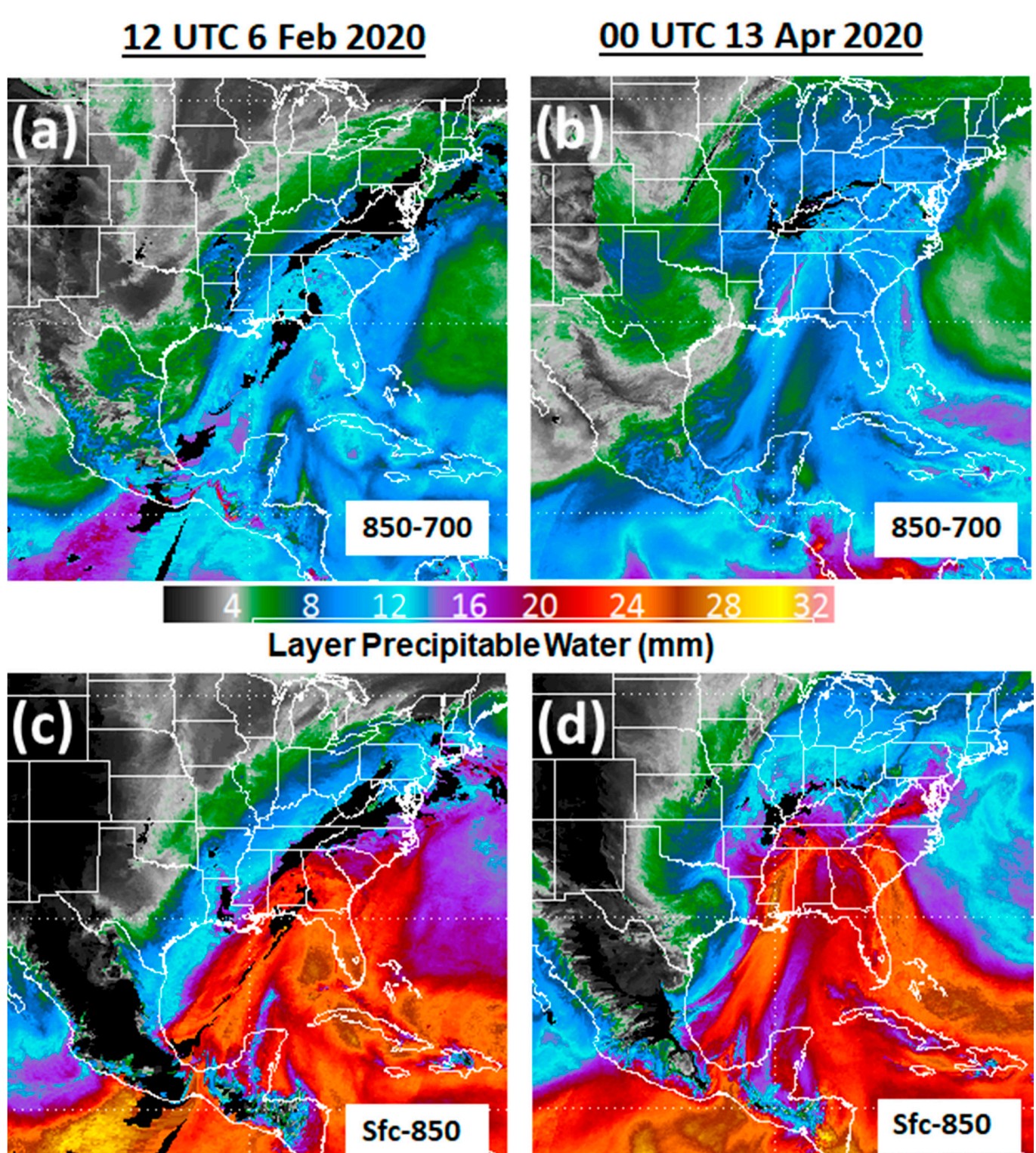

**Figure 11.** Advected Layer Precipitable Water (ALPW, mm) for the 850-700 hPa and surface-850 layers of the (**a**,**c**) February and (**b**,**d**) April 2020 case.

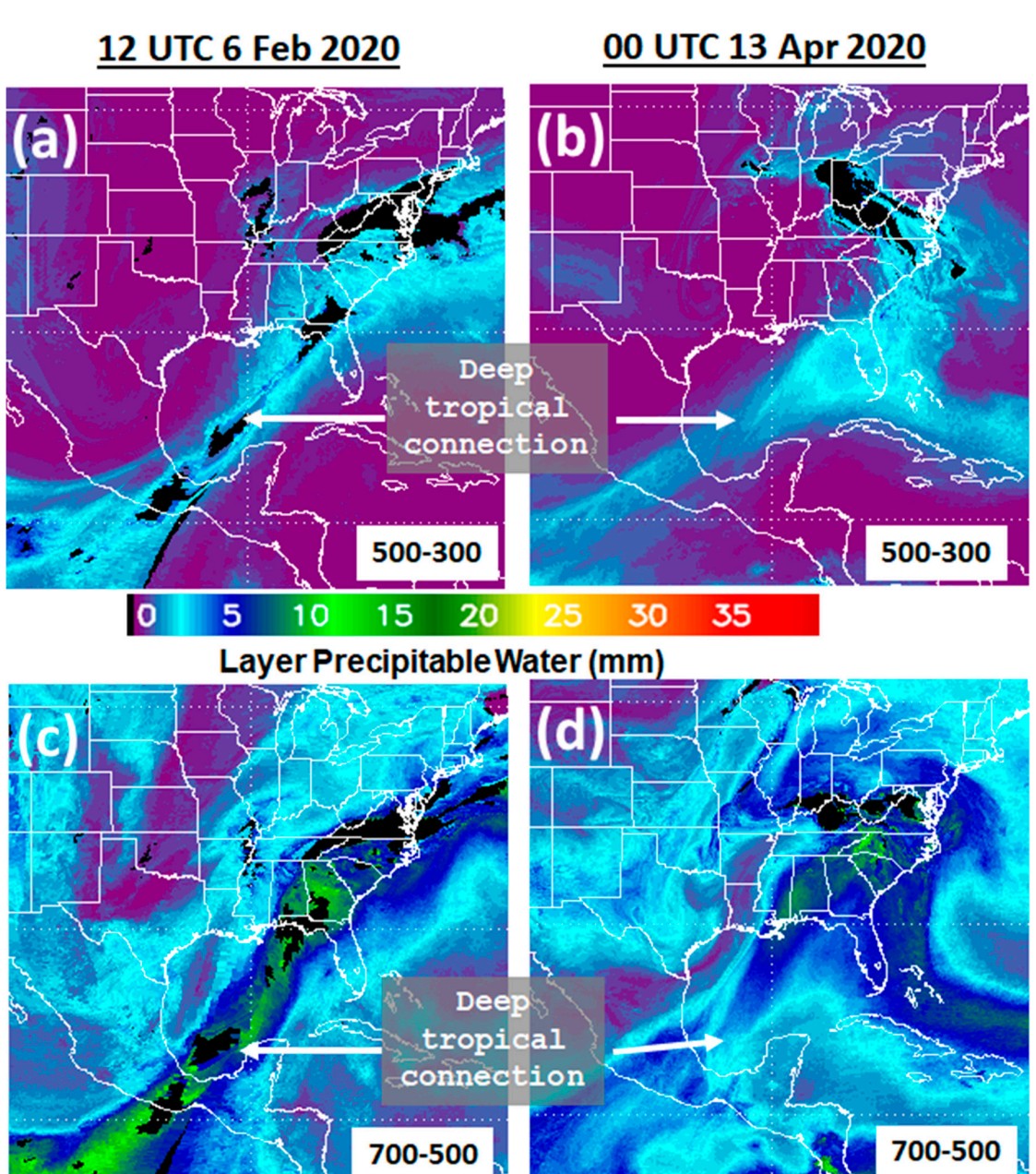

**Figure 12.** As in Figure 11, but for the (**a**,**b**) 500-300 and (**c**,**d**) 700-500 hPa layers.

### 3.3. Rainfall Observations

As rainfall events can serve to condition and/or trigger landslides, a broad view of rainfall affecting the PRB and CRB is warranted. A time series of bi-monthly average per gauge accumulation based on an 11-year climatology of Duke GSMRGN observations in the PRB and observed amounts for the fall 2019–spring 2020 period are plotted in Figure 13a. Additionally, occurrences of extreme (top 2.5%) rain events (ExtR) and Elevated Rain Time Clusters (Ext ERTCs) of the PRB during the fall 2019–spring 2020 period are plotted. Bi-monthly data points are plotted on the 7th and 22nd day of each month and correspond to the collection of rainfall events during the first half (1–15) and second half (16–end of month) days of each month. After the heightened rainfall activity associated with the remnants of tropical systems Nestor and Olga in October 2019, there was a prolonged "quiet" period in late 2019 when observed rain events tracked climatology in both basins (Figure 13a,b; note differences in the range of the ordinate axes). Late January 2020 rainfall

amounts dropped below climatology, even in the midst of an Ext ERTC in the PRB that occurred from 21 January–16 February 2020. This particular ERTC qualified as extreme due primarily to the ExtR 5–7 February 2020 event falling within the ERTC period. This event was also responsible for the above-normal rainfall "spike" of early February 2020 in both basins. Rainfall observed by the Duke GSMRGN in late February and early March 2020 dropped below "normal" before the occurrence of the ExtR on 20–25 March 2020 and the Ext ERTC from 28 March–9 April 2020. Hence, there was an extended period of significant rainfall in the PRB from 20 March–9 April, just before the ExtR 12–13 April 2020 event.

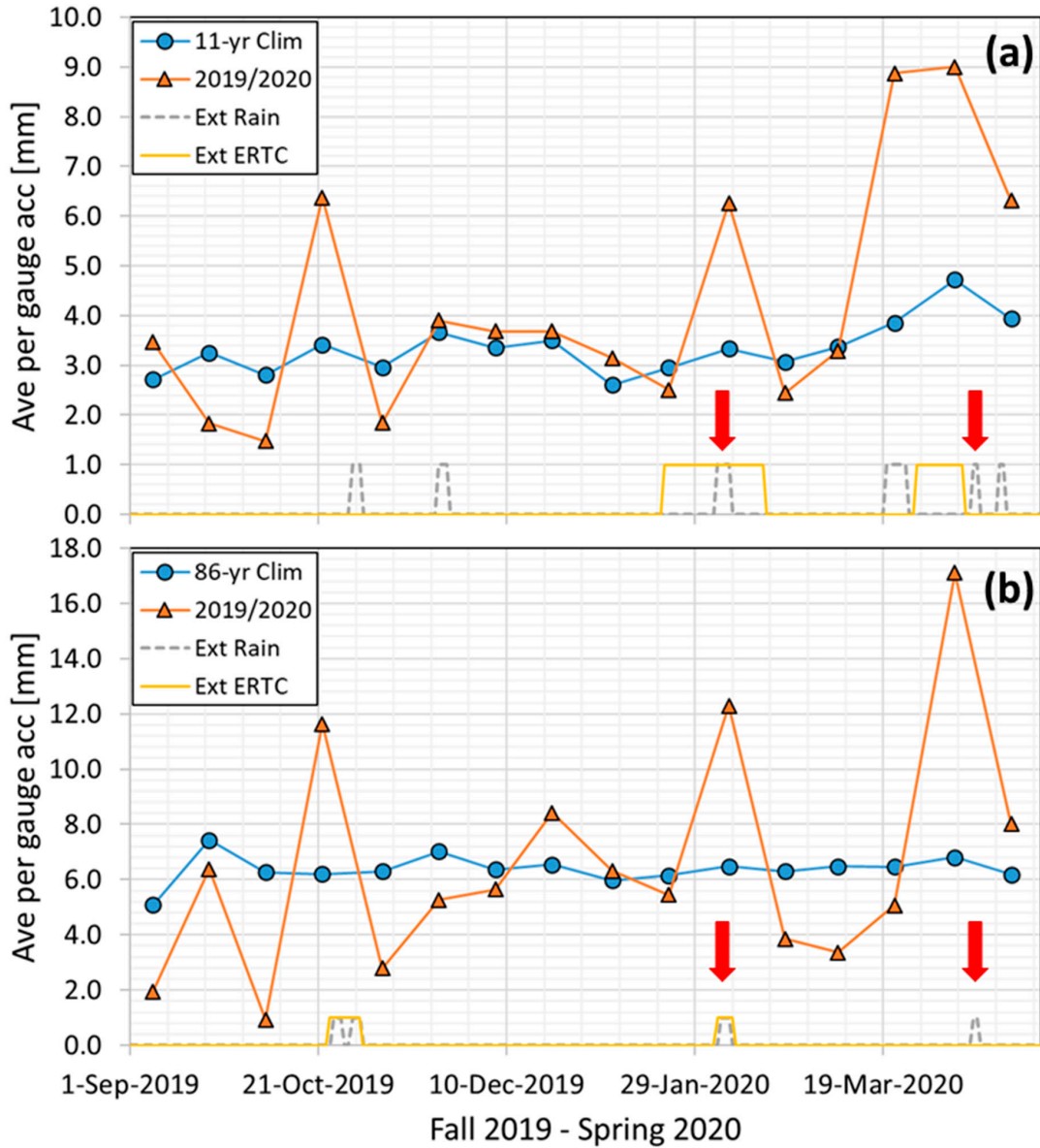

**Figure 13.** Bi-monthly average per gauge accumulation (mm) of the: (**a**) Duke GSMRGN located in the PRB; (11-year) and (**b**) CHLRGN located in the CRB (86-year) climatology (blue circles) and observed amounts for fall 2019-spring 2020 (orange triangles). Occurrence of extreme (top 2.5%) rain events (ExtR) and Elevated Rain Time Clusters (Ext ERTCs) of the watersheds over the period plotted at the bottom of the panels as dashed grey and solid gold lines, respectively. The focus of this study is on the two ExtR events (5–7 February and 12–13 April 2020) highlighted in the panels with a red arrow. Note the difference in range of the ordinate axis between the panels.

From mid-March until the 12–13 April event, the ExtR and second Ext ERTC events of the PRB (Figure 13a) were influenced by ARs and "bursts" of relatively high southerly IVT events not qualifying as ARs (hereafter referred to as atmospheric "creeks", not shown).

The 20–25 March ExtR was influenced by an AR whose storm center tracked toward the northeast during its first half and an atmospheric creek that tracked south of North Carolina in the second half. The second Ext ERTC event (28 March–9 April 2020) originated during a period of prolonged warm air advection as a decaying AR approached the PRB from western Tennessee and its associated cyclone center tracked toward the northeastern U.S. (not shown). Pulses of shortwaves moving southeastward brought cyclonic vorticity advection, lift, and precipitation to the southern Appalachians in early April until just after a convective line of precipitation associated with a cold front passed through the PRB and CRB early in the morning of 9 April 2020.

Comparison of the event rainfall accumulation at each available gauge of the Duke GSMRGN in the PRB for the two ExtR events of 5–7 February and 12–13 April 2020 is plotted in Figure 14. The 5–7 February accumulated rainfall amounts were significantly higher than for the 12–13 April amounts. The February event was influenced by two ARs extending over a rainfall period of 54 h. The accumulation amounts 178, 162, and 179 mm (Figure 14a) along the ridgelines immediately northwest of Cataloochee Creek (red star in Figure 14) fed directly into the creek and contributed to its significant flooding and the wash-out of a nearby road. Although the net accumulation during the April event was smaller than that of the February event, the mean event rain rate was higher, as the event period was only 30 h. Higher rain rates observed during the April event are consistent with the greater convective influence on precipitation during this event and will be explored in greater detail in a follow-on paper to this study. As in the February event, the accumulation amounts 101, 81, and 132 mm during the April event (Figure 14b) along the ridgelines immediately northwest of Cataloochee Creek contributed additional flooding and wash-out-related damage to the nearby road in the GSMNP. The event accumulation mean (standard deviation) of the two events in the PRB was 138 mm (25 mm) and 94 mm (25 mm), respectively. Comparable values of the two events in the CRB (not shown), as measured by the CHLRGN, were 157 mm (8 mm) and 126 mm (23 mm), respectively. Recall the CRB area and topographic variation is relatively small (Appendix A), so a standard deviation of 23 mm for the second event suggests that localized (convective) rainfall observed by the CHLRGN was substantial. Furthermore, it is noteworthy that the CRB is located in close proximity to the BRE, a source of broad and significant orographically enhanced lift, so that larger-scale atmospheric disturbances generally deposit higher rainfall amounts in the CRB than in the PRB (compare the bi-monthly climatological (blue) traces in Figure 13a,b).

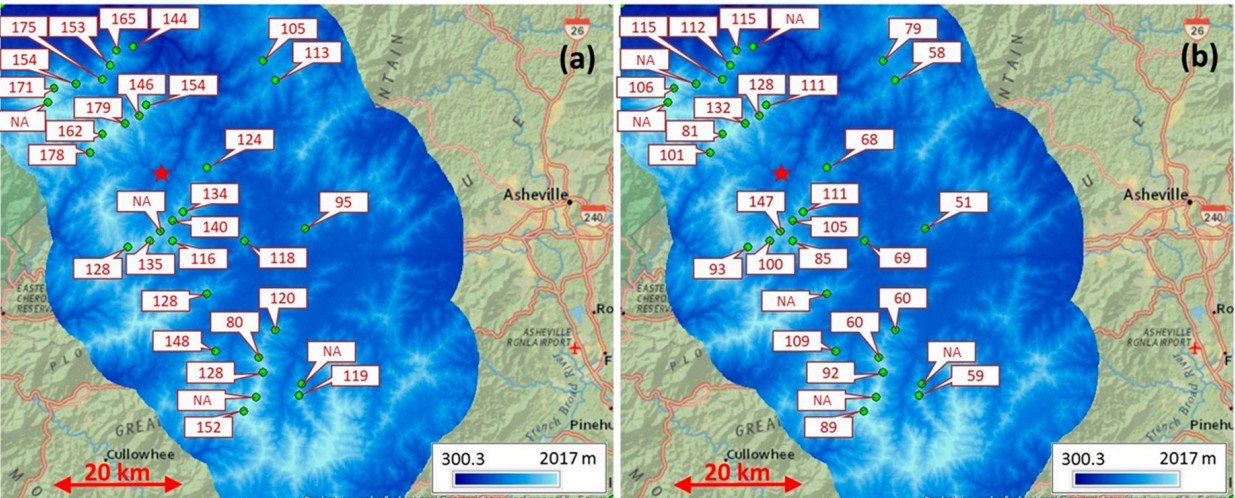

**Figure 14.** Rainfall accumulation (mm) observed by the Duke GSMRGN in the PRB over the period: (**a**) 5–7 February, and (**b**) 12–13 April 2020. Flag "NA" indicates rain gauge was not reported during the period. Red stars highlight the location of Cataloochee Creek in the GSMNP.

Historical Perspective

Given the 11-year history of observations by the Duke GSMRGN, it is noteworthy that the rankings of the February and April 2020 events were 4 and 21, respectively, of the 2,321 events registered by the gauge network up until 30 June 2020. These rankings easily qualified as ExtR events (top 0.17% and 0.90%, respectively). The ERTCs associated with both events as observed by the Duke GSMRGN also qualified as extreme as the rankings of the 21 January–16 February and the 28 March–9 April events were 5 and 6 (top 0.97% and 1.2%), respectively, out of 516 ERTCs observed during the 11-year period through 30 June 2020.

For context, the longer-period 86-year CHLRGN record of rainfall events (Figure 13b) ranked the two individual AR-influenced events of early 2020 as 60 and 118 (top 0.60% and 1.2%), respectively, of the 10,012 events registered by the gauge network up until 30 June 2020, easily qualifying them as ExtR events. The ERTC event rankings differed slightly from those based on observations of the PRB (Figure 13a). The ERTCs associated with both events as observed by the CHLRGN had a significantly shorter time period (4–8 February and 12–13 April, respectively), with rankings of 90 and 203 (top 1.7% and 3.8%), respectively, out of the 5320 ERTCs observed during the 85-year period through 30 June 2020. Hence, the second ERTC merely qualified as a strong (top 5.0%) rather than an extreme event. This ranking difference could be a result of differences in period of record between the two datasets. However, it is equally possible that rankings differences arose from differences in areal coverage of each gauge network. The CHLRGN covers a much smaller area and its rainfall database is more sensitive to mid-latitude storm tracks and the positioning of their associated ARs or atmospheric creeks. Proximity of the CRB to the BRE is also a key contributor to differences in observed storm accumulation and rainfall severity ranking between the two basins during passage of tropical storm Olga in late October 2019 and of the two heavy rainfall events in early 2020 (Figure 13a,b). A large-scale atmospheric disturbance must exceed a higher threshold to reach the extreme (top 2.5%) precipitation category in the CRB since large-scale precipitation events locally have additional lift provided by the broad orography of the BRE.

## 4. Discussion

Miller et al. [38] examined landslide events in the southern Appalachian Mountains from the perspective of conditioning of the soil before the onset of precipitation events associated with them. It was found that long periods of rainfall (extreme elevated rain time clusters, Ext ERTCs) often linked with individual extreme rainfall events (ExtR) and sometimes with ARs, showed a relatively high correlation with landslide days occurring within 30 days after the rainfall event (Pearson correlation coefficient of 0.561 and $p$-value of 0.008 for 117 data pairs). The April 2020 event of this study showed that important soil conditioning before the onset of a landslide-triggering rain event could extend beyond the 30-day lag period utilized in Miller et al. [38] and highlighted in recent studies [39–41]. Twenty-one landslides were conditioned and triggered during the 12–13 April event over a broad region near the CRB. The landslides triggered by both events, along with post-event flooding as estimated by the VIIRS/ABI algorithm and post-event shallow moisture drying as estimated by the SMOPS algorithm were consistent with the soil water storage capacity having not been (having been) exceeded during the February (April) 2020 heavy rainfall event. The lack of extreme rainfall in the CRB over the interim period between 7 February and 12 April 2020 (Figure 13b) suggests that the February event pre-conditioned the soil in and near the CRB such that the antecedent moisture, coupled with the early stratiform precipitation of the April event, was sufficient to serve as the cause of the 21 landslides, with the convective rainfall of the April event serving as their trigger. The interaction of these various factors will be a focus of the study presented in a follow-on paper. The interim period between the two heavy rainfall events in early 2020 exceeded the 30 day lag limit used in the Miller et al. [38] study that focused only on extreme rainfall events as pre-conditioners of landslide initiation. Another focus of the follow-on paper to the study

will be to utilize precipitation anomalies (departures from climatology) in a given river basin (Figure 13), including all rainfall events (from ordinary to extreme), to investigate the integrated anomaly as a proxy for relative mid- and lower-layer (deep) soil moisture saturation in the basin.

The landslide near Bunches Bald, triggered by the February 2020 storm, was an isolated anomaly in which the antecedent soil moisture and total event rain accumulation and rain rate intensity were insufficient to force widespread landslides as was observed in April 2020. The NCGS survey of the February 2020 landslide in the PRB showed the initiation point to be on a modified slope. Wooten et al. [74] showed that threshold peak rain rates for triggering landslides are lower for modified than for unmodified slopes. Hence, under favorable atmospheric scenarios, precipitation falling during a single rain event is sometimes sufficient to condition the soil and trigger a shallow landslide on a modified slope, as was observed during the February 2020 event. Another factor related to the vulnerability of the soil at the landslide initiation point near Bunches Bald was the influence on soil composition and surface water flow by material from an earlier landslide at the same location initiated in April 2019 [75].

## 5. Conclusions

The heavy rainfall events of early 2020 caused significant damage to infrastructure such as eroding roadways in the Great Smoky Mountains National Park [67] and in a broad region of the southern Appalachian Mountains. The large-scale weather pattern responsible for the February and April 2020 extreme rainfall events, as identified in Miller et al. [46], started with a hang-back trough that, with its associated slow propagation, permitted ample time for low-level air streams to be humidified and heated by the warm ocean surfaces of the tropics and sub-tropics. The air streams circulated such that they were drawn poleward by a developing mid-latitude cyclone and located just ahead of the associated cold front, a feature known as an atmospheric river (AR). Each event was associated with at least one AR. The February 2020 event was associated with two during its longer duration (54 h). Both events were preceded by potential vorticity structures consistent with anticyclonic Rossby wave breaking in the upper-troposphere/lower-stratosphere, a key finding from the climatology of Moore et al. [70] in which the resulting high amplitude large-scale weather pattern more often led to extreme precipitation events.

The long duration February 2020 event produced significantly more rainfall accumulation observed by gauge networks in the PRB and CRB; however, the event hourly rain rates were rather modest compared to those observed during the shorter-duration (30 h) April 2020 event. Thus, the latter event accumulated a larger fraction of event rainfall during the passage of convective elements. The greater number of thunderstorm wind and tornado reports documented by the National Weather Service during the April event confirmed the significant contribution of convective elements. Differences in peak hourly rain accumulations observed by the two gauge networks appear linked to the post-event surface response in the form of landslides. Although the February 2020 event total accumulation was greater than that of the April 2020 event, its lighter rain rates resulted in only two landslides documented by scientists of the North Carolina Geological Survey compared with 21 landslides of the April 2020 event. The accumulated rainfall occurring during both events qualified as extreme (top 2.5%, ExtR) in the data records of the PRB and CRB gauge networks and are often associated in the cool season with mid-latitude storms having ARs [46].

Space-borne observations offered broad areal and temporal contexts of the two events not possible with in situ instrumentation in the southern Appalachians, such as rain gauges. Layered precipitable water (ALPW) observations indicated a deep layer of significant moisture associated with the AR of the February 2020 event, extending from the surface to the 500 hPa level. ALPW observations during the April 2020 event showed significant water vapor from the surface to the 700 hPa level, but relatively smaller amounts above the 700 hPa level compared to observations of the February event. Gridded GFS analyses of

mixing ratio and $\theta e$ at the 600 and 850 hPa levels during the April event are consistent with these observed differences and are also consistent with a greater contribution by convective elements in the April event due to the release of convective instability ($\delta\theta e/\delta Z < 0$) by large-scale lift downstream of the broad 500 hPa cyclone. Shallow soil moisture (SMOPS) observations immediately after the two events indicated a more rapid shallow soil moisture drop-off after the February event compared to after the April event. Post-event soil moisture decline after the April storm occurred over an extended period of three days and was interpreted as the consequence of reduced water runoff and/or percolation rates from the upper-soil layer due to the saturated mid- and lower-soil layers. Observations over a rectangular region stretching between Newton and Chattanooga, Tennessee of flooded pixels (VIIRS/ABI) reflected the unique post-event landslide and soil moisture responses of the two events. Downstream effects as quantified by the percentage and duration of daily flooded-to-available pixels in the region were significantly smaller after the February event compared to the immediate post-April event period. SMOPS and VIIRS/ABI observations and the expanse of triggered landslides after passage of the two events imply that the mid- and lower (deep)-soil layers were unsaturated (but had elevated moisture amounts) after the February storm and surpassed saturation (storage capacity) after the April storm. From the perspective of surface conditions, antecedent soil moisture from the February 2020 event was the cause of the post-April event landslides, after the soil was additionally conditioned and triggered by rainfall during passage of the April storm. Hence, the aftereffects of the April 2020 event require a broadening of operational forecast considerations beyond the 30 day lag pre-conditioning focus of Miller et al. [38].

**Author Contributions:** Conceptualization, D.M. and R.F.; methodology, D.M., J.F., S.K., W.S.III, J.Y., X.Z. and R.F.; formal Analysis, D.M., J.F., S.K., W.S.III, J.Y. and X.Z.; data Curation, D.M., J.F., W.S.III and X.Z.; writing—original draft preparation, D.M.; writing—review and editing, J.F., S.K. and R.F. All authors have read and agreed to the published version of the manuscript.

**Funding:** Maintenance costs of the Duke GSMRGN were supported by NOAA through the Cooperative Institute for Satellite Earth System Studies under Cooperative Agreement NA19NES4320002. NASA grants NNX07AK40G, NNX10AH66G, and NNX13AH39G covered installation and original maintenance costs of the Duke GSMRGN to Ana Barros at Duke University.

**Institutional Review Board Statement:** Not applicable.

**Informed Consent Statement:** Not applicable.

**Data Availability Statement:** Data sets utilized in this study not having an appropriate URL listing in the manuscript are available upon request to the lead author.

**Acknowledgments:** We gratefully acknowledge the NOAA GOES-R and JPSS programs for providing support to the satellite algorithms and the Duke GSMRGN. The authors are grateful for the support and assistance of Paul Super and Tom Remaley of the National Park Service, land owners who have permitted the installation of a rain gauge on their property, University of North Carolina at Asheville and Duke University students, and Kyle, Don, Hugh, and Roger, support personnel of the Waynesville Watershed. We are grateful for the numerous discussions with Rick Wooten and Corey Scheip of the North Carolina Geological Survey. We also acknowledge the collection and sharing of rainfall observations by A. Christopher Oishi and Patsy Clinton of the USDA Forest Service, Coweeta Hydrologic Laboratory contained in their data archive. We also gratefully acknowledge the helpful comments of several anonymous reviewers who greatly improved the quality of the manuscript. Color scales used in figures having shading were designed by Cynthia Brewer and Mark Harrower (http://colorbrewer2.org, accessed on 12 December 2020). Additionally, appreciation of Shawn Milrad (http://www.shawnmilrad.com/gempak-color-tables-and-stuff, accessed on 12 December 2020) must be noted whose instructions allowed the pleasing color tables of "colorbrewer" to be incorporated into the shading of GEMPAK-generated figures.

**Conflicts of Interest:** The manuscript contents are solely the opinions of the authors and do not constitute a statement of policy, decision, or position on behalf of NOAA or the U. S. Government. The authors declare no conflict of interest. The funders had no role in the design of the study; in the collection, analyses, or interpretation of data; in the writing of the manuscript, or in the decision to publish the results.

## Appendix A

**Table A1.** Location and elevation of the 32 tipping bucket rain gauges comprising the Duke GSMRGN and of the nine NOAH IV weighing rain gauges (WRGs) comprising the CHLRGN.

| Duke GSMRGN Gauge Attributes | | | | | | | | CHLRGN Gauge Attributes | | | |
|---|---|---|---|---|---|---|---|---|---|---|---|
| Gauge | Lat. | Lon. | Elev. (m) | Gauge | Lat. | Lon. | Elev. (m) | Gauge | Lat. | Lon. | Elev. (m) |
| RG002 | 35°25.5′ | 82°58.2′ | 1731 | RG109 | 35°29.7′ | 83°02.4′ | 1500 | WRG06 | 35°3.62′ | 83°25.8′ | 687 |
| RG003 | 35°23.0′ | 82°54.9′ | 1609 | RG110 | 35°32.8′ | 83°08.8′ | 1563 | WRG05 | 35°3.63′ | 83°27.9′ | 1144 |
| RG004 | 35°22.0′ | 82°59.4′ | 1922 | RG111 | 35°43.7′ | 82°56.8′ | 1394 | WRG20 | 35°3.89′ | 83°26.5′ | 740 |
| RG005 | 35°24.5′ | 82°57.8′ | 1520 | RG112 | 35°45.0′ | 82°57.8′ | 1184 | WRG31 | 35°1.96′ | 83°28.1′ | 1366 |
| RG008 | 35°22.9′ | 82°58.4′ | 1737 | RG300 | 35°43.5′ | 83°13.0′ | 1558 | WRG13 | 35°3.75′ | 83°27.4′ | 961 |
| RG010 | 35°27.3′ | 82°56.8′ | 1478 | RG301 | 35°42.3′ | 83°15.3′ | 2003 | WRG41 | 35°3.32′ | 83°25.7′ | 776 |
| RG011 | 35°23.7′ | 82°54.9′ | 1244 | RG302 | 35°43.2′ | 83°14.8′ | 1860 | WRG12 | 35°2.84′ | 83°27.5′ | 1001 |
| RG100 | 35°35.1′ | 83°04.3′ | 1495 | RG303 | 35°45.7′ | 83°09.7′ | 1490 | WRG55 | 35°2.39′ | 83°27.3′ | 1035 |
| RG101 | 35°34.5′ | 83°05.2′ | 1520 | RG304 | 35°40.2′ | 83°10.9′ | 1820 | WRG96 | 35°2.72′ | 83°26.2′ | 894 |
| RG102 | 35°33.8′ | 83°06.2′ | 1635 | RG305 | 35°41.4′ | 83°07.9′ | 1630 | | | | |
| RG103 | 35°33.2′ | 83°07.0′ | 1688 | RG306 | 35°44.7′ | 83°10.2′ | 1536 | | | | |
| RG104 | 35°33.2′ | 83°05.2′ | 1587 | RG307 | 35°39.0′ | 83°11.9′ | 1624 | | | | |
| RG105 | 35°38.0′ | 83°02.4′ | 1345 | RG308 | 35°43.8′ | 83°10.9′ | 1471 | | | | |
| RG106 | 35°25.9′ | 83°01.7′ | 1210 | RG309 | 35°40.9′ | 83°09.0′ | 1604 | | | | |
| RG107 | 35°34.0′ | 82°54.4′ | 1359 | RG310 | 35°42.1′ | 83°07.3′ | 1756 | | | | |
| RG108 | 35°33.2′ | 82°59.3′ | 1277 | RG311 | 35°45.9′ | 83°08.4′ | 1036 | | | | |

## Appendix B

**Table A2.** List of uncommon abbreviations used in describing the study.

| Abbreviation | Definition |
|---|---|
| AR | Atmospheric River |
| BRE | Blue Ridge Escarpment |
| CHLRGN | Coweeta Hydrologic Laboratory Rain Gauge Network |
| CRB | Coweeta River Sub-Basin |
| Duke GSMRGN | Duke Great Smoky Mountains Rain Gauge Network |
| ERTC | Elevated Rain Time Cluster |
| ExtR | Extreme (top 2.5%) Rainfall event |
| PRB | Pigeon River Basin |
| ULTRB | Upper Little Tennessee River Basin |

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
