# Peer review of "A Study of Two Impactful Heavy Rainfall Events in the Southern Appalachian Mountains during Early 2020, Part I; Societal Impacts, Synoptic Overview, and Historical Context"

_remotesensing, doi:10.3390/rs13132452_

Round 1
Reviewer 1 Report
Manuscript has several instances of plagiarism; I recommend changing the text and resubmit the manuscript.
- Line 98-108 from https://www.mdpi.com/2073-4433/10/2/71
- Line 129-142 from https://www.mdpi.com/2073-4433/10/2/71
- Line 148-152 from https://www.mdpi.com/2073-4433/10/2/71
- Line 160-172 from https://www.mdpi.com/2073-4433/10/2/71
- Line 214-220 from https://www.mdpi.com/2073-4433/10/2/71
Author Response
Please find my response to reviewer comments in the attached document.

Reviewer 2 Report
The topic is interesting. However, I think that it cannot be published in its present form. I suggest that part I and II could merge and submitted as a common paper. This paper is very descriptive for the standards of the journal and lacks dynamic interpretation. What are the mechanisms responsible for the two case studies? what differentiates the one from the other? Furthermore, I got confused with the climatology of the events since some discussion is based on the climatology and previous studies and some others on the two cases.
Author Response

(The authors gave the same response as above.)

Reviewer 3 Report
The article entitled “A Study of Two Impactful Heavy Rainfall Events in the Southern Appalachian Mountains during Early 2020, Part I; Societal Impacts, Synoptic Overview, and Historical Context” describes an extensive analysis of the consequences of two heavy rainfall events in an area of the USA. The study is well developed and the content is overall interesting. The manuscript is in general well written and presented in a good form. However, some issues are present according to this Reviewer. These issues could be addressed with a revision. All details are indicated in the following.
Required changes:
- Section 2.3.1: the Authors state in this section that satellites can provide measurements of the changes of soil moisture. The description contained in this section is not convincing. A comparison between the soil moisture measured with the procedure described by the Authors and the values measured employing more traditional instruments should be presented.
- Section 3.1.2: contrarily to the previous section (about flooding), this one is too short. In particular, the Authors should report more information concerning the considered landslides, such as for example if they are shallow or deep landslides, the volume of soil involved and so on.
- Figure 9: according to the Authors, several landslides occurred in the considered period of observation. Fig. 9 shows the variation of soil moisture with time, referring to a large area. However, the soil moisture is very variable in space, and it should be a punctual measurement rather than referred to a large area. Moreover, it also varies with depth.
- Lines 71-73: this concept is interesting. However, some references could be added. For the sake of completeness, some suggested references are reported in the following.
SUGGESTED REFERENCES
Conte, E.; Pugliese, L.; Troncone, A. (2020). Post-failure analysis of the Maierato landslide using the material point method. Eng. Geol., 277, 105788.
Author Response

(The authors gave the same response as above.)

Reviewer 4 Report
1. Line 98: Please briefly introduce the characteristics of your study sites in Pigeon River Basin (PRB) and Coweeta River Basin (CRB) in Figure 1; ex. areas, slope, elevation, land use/ land cover, etc. It is helpful for reader to understand your study sites. 2. Line 149: The geodatabase documents 23 landslides. What kind of landslide types? How about their areas and amounts of landslide movement? 3. Line 192: The SMOPS V3.0 has higher accuracy of soil moisture. How about the accuracy comparing with the real surface observations? 4. Line 240: There were two heavy rainfalls of the 5-7 February and 12-13 April 2020 events. Please show the histogram and hydrograph figures of these two rainfall events at specific sites. 5. Line 331-418: Figures 2 - 8 are clearly maps which are helpful for reader to understand. They are excellent works. The descriptions for each figure are also detailed. 6. Line 442: “The April case had a better-developed source from the Atlantic Ocean, but the February event had a deeper moist layer than the April event, with slightly higher values above 700 hPa.” Please explain this phenomenon more detail. 7. Line 525: Based on historical perspective, the 11-year history of observations by the Duke GSMRGN is too short. So, the ranking of the February and April 2020 events was 4 and 21, respectively.Author Response
Please find my response to reviewer comments in the attached document.

Round 2
Reviewer 1 Report
Thank you for considering the comments and adding the text to the revised version.
Author Response
Please find my response to reviewer comments concerning the second version of the manuscript in the attached document.

Reviewer 2 Report
The authors have made a great effort to improve the manuscript considering the comments of the reviewers. I think that the title is not appropriate since there is no reason to note "Part I" and "Part II". Similarly there is no need to mention Part I and ParII in the introduction (lines 80-88). It is a paper that includes the analysis of two high impact cases. If the authors would like to write a second paper for the same cases they should find another title to describe the scope and results of the second paper.. They should not refer to the future paper as Part II. In the end of the conclusions they could refer to their future plans and a forthcoming paper.
In addition, I think that the abstract is very descriptive without incorporating main conclusions. The reference to another paper in the abstract in general is not accepted. Furthermore, the distinction between PartI and Part II is also mentioned in the abstract. This is not what the reader would like to know from the abstract of a paper.
Author Response

(The authors gave the same response as above.)

Reviewer 3 Report
The issues raised during the first revision have been addressed.
Author Response

(The authors gave the same response as above.)
